# Strong neuron-to-body coupling implies weak neuron-to-neuron coupling in motor cortex

Patrick A. Kells[1], Shree Hari Gautam [1], Leila Fakhraei[1], Jingwen Li[1] & Woodrow L. Shew [1]

Cortical neurons can be strongly or weakly coupled to the network in which they are embedded, firing in sync with the majority or firing independently. Both these scenarios have potential computational advantages in motor cortex. Commands to the body might be more robustly conveyed by a strongly coupled population, whereas a motor code with greater information capacity could be implemented by neurons that fire more independently. Which of these scenarios prevails? Here we measure neuron-to-body coupling and neuron-to-population coupling for neurons in motor cortex of freely moving rats. We find that neurons with high and low population coupling coexist, and that population coupling was tunable by manipulating inhibitory signaling. Importantly, neurons with different population coupling tend to serve different functional roles. Those with strong population coupling are not involved with body movement. In contrast, neurons with high neuron-to-body coupling are weakly coupled to other neurons in the cortical population.

---

[1] Department of Physics, University of Arkansas, Fayetteville, Arkansas 72701, USA. Correspondence and requests for materials should be addressed to W.L.S. (email: shew@uark.edu)

The functions carried out by the cerebral cortex require coordinated interactions among large networks of neurons. Whether these interactions result in a state of collective cortical dynamics that is synchronous or asynchronous has been debated for decades[1–7]. This debate stems from examining the system at two different levels. At the macroscopic network level, measurements of local field potential or the summed spiking activity of a large population often reveal prominent oscillations and synchrony[8–11]. In contrast, at the single-cell level, it is often found that the spikes of any two cortical neurons are rather independent of each other, suggesting an asynchronous, weakly coupled cortical state (particularly in alert, awake animals)[12–14]. One strategy for reconciling these dichotomous viewpoints comes from bridging the two levels, i.e., directly comparing single neuron spiking to the collective activity of the network in which the neuron is embedded. With this approach, it was recently shown that there is remarkable diversity among neurons; some neurons are strongly coupled to collective population spiking activity, whereas others are more independent[15]. Similarly, the strength of correlations between single neurons and local field potential can be diverse[15–20]. These facts suggest a revision of the old debate. Instead of asking whether the cortical network is strongly or weakly coupled, we should acknowledge that the cortex is composed of both strongly and weakly coupled neurons, and turn to a new question. Do the neurons with weak population coupling have different roles in cortical function than neurons with strong population coupling?

Several theoretical arguments suggest that differing degrees of population coupling could be functionally important. For instance, if a given cortical function is executed by a strongly coupled population, then the function could benefit from "strength in numbers," gaining robustness well-suited to coherent signaling across distant cortical regions[9,21]. On the other hand, strong population coupling could manifest as fluctuations unrelated to the relevant signal, i.e., noise correlations, which can undermine signal-to-noise[22,23]. Moreover, weaker population coupling could be beneficial, as weaker correlations can enhance the information capacity of population coding[24–26]. Recent work in visual cortex demonstrated that neurons with high population coupling exhibit stronger response to visual input, suggesting that some aspects of visual coding may leverage the robustness of strong population coupling[15]. Do similar functional roles of neurons with weak and strong population coupling exist in motor cortex? Although previous studies demonstrate that correlations among single units can have a role in motor coding (e.g., refs. [27,28]), and that some neurons in motor cortex appear to be more strongly coupled to the population than others[20,29], it remains unclear whether motor coding may differ across neurons with strong vs. weak population coupling. Here we address this question in motor cortex of freely moving awake rats. Our primary finding is that neurons that were strongly coupled to the population were weakly related to body movement. Conversely, neurons with strong coupling to the body exhibited weak population coupling.

## Results

**Inhibitory modulation of voluntary movement and motor cortex.** We performed microelectrode array recordings of neural activity simultaneously with high-precision measurements of the rats' body movements (Fig. 1a). Body movement was captured with 10 ms temporal resolution and submillimeter spatial resolution using nine infrared cameras to track the three-dimensional motion of eight small reflective beads attached to the rat's head, back, and tail. We first analyzed the body speed, averaged over all eight beads (Fig. 1b). Unlike typical studies of the motor cortex,

this experimental system allowed us to study the unconstrained and untrained natural voluntary movements of the rats. Studying such natural voluntary movements is likely to be particularly important in the context of brain disorders with abnormal repertoires of voluntary movement, such as autism. Using 32-channel electrode arrays chronically implanted in deep layers of motor cortex (600–1200 μm), we obtained single-unit spiking activity (Fig. 1b, Supplementary Fig 1) ($n = 1258$ single units, $n = 143$ recordings, 30 min each, $n = 6$ rats). Considering that consecutive recordings from the same rat are likely to entail repeated measurements of many of the same units, a lower bound on the number of unique units we recorded is 119. The spatial location and extent of the electrode arrays was chosen such that the recorded units are likely to be associated with movement of many parts of body including the whiskers, neck, trunk, hips, wrists, and more. Thus, we sought general relationships between motor cortical neurons and body movement rather than detailed coding strategies of specific motor tasks.

To obtain a broader range of behavioral and neural states, we compared normal rats with those with pharmacologically altered inhibitory synaptic signaling. Low doses of $GABA_A$ antagonist or agonist were administered to manipulate inhibition. In one cohort of animals (group 1, $n = 3$ rats), we studied systemic changes in inhibition (pentylenetetrazol (PTZ) at 30 mg/kg intraperitoneally (IP) or muscimol at 2 mg/kg IP). In group 1, we performed 31 no drug recordings, 18 muscimol recordings, and 16 PTZ recordings. In another cohort (group 2, $n = 3$ rats), we studied local changes in inhibition employing local drug infusion in motor cortex (20–1280 μM of bicuculline or muscimol). In group 2, we performed 38 no drug recordings, 19 muscimol recordings, and 21 bicuculline recordings. We anticipated that such alterations of the balance of excitation and inhibition would alter the collective population activity of motor cortex, thus changing population coupling. By controlling population coupling in this way, we aimed to more thoroughly explore the link between population coupling and motor function.

Before examining changes in population coupling, we first show how our manipulations of inhibition affected some basic aspects of body movement and spike rates in the motor cortex. First, we found that systemically suppressed inhibition was correlated with increased voluntary animal movement (Fig. 1c, Spearman's $\rho = 0.73$, $p < 10^{-4}$), whereas systemically enhanced inhibition decreased movement ($\rho = -0.46$, $p < 10^{-3}$). (We note that here and throughout the text the $p$-values were determined based on tests described in the Online Methods section). Spike rates were not significantly correlated with these systemic inhibitory manipulations (Fig. 1d). In contrast, the concentration of locally applied muscimol was positively correlated with body movement (Fig. 1c, $\rho = 0.49$, $p < 10^{-3}$) and anticorrelated with spike rates (Fig. 1d, $\rho = -0.15$, $p < 0.02$). Local bicuculline concentration was not significantly correlated with either firing rates or body movement.

**Inhibitory modulation of population coupling in the motor cortex.** Next we assessed how the firing of each single unit was related to the collective activity of the network in which it was embedded; we computed the population coupling of each unit[15,30]. In brief, population coupling quantifies how the spike count time series of one unit co-varies with the summed spike count time series of the rest of the recorded population (Methods). As observed previously in the sensory cortex[15] and motor cortex[29], we observed that some neurons were strongly coupled to the population, while others fired more independently (Fig. 2a). Population coupling was most often positive (correlated with the

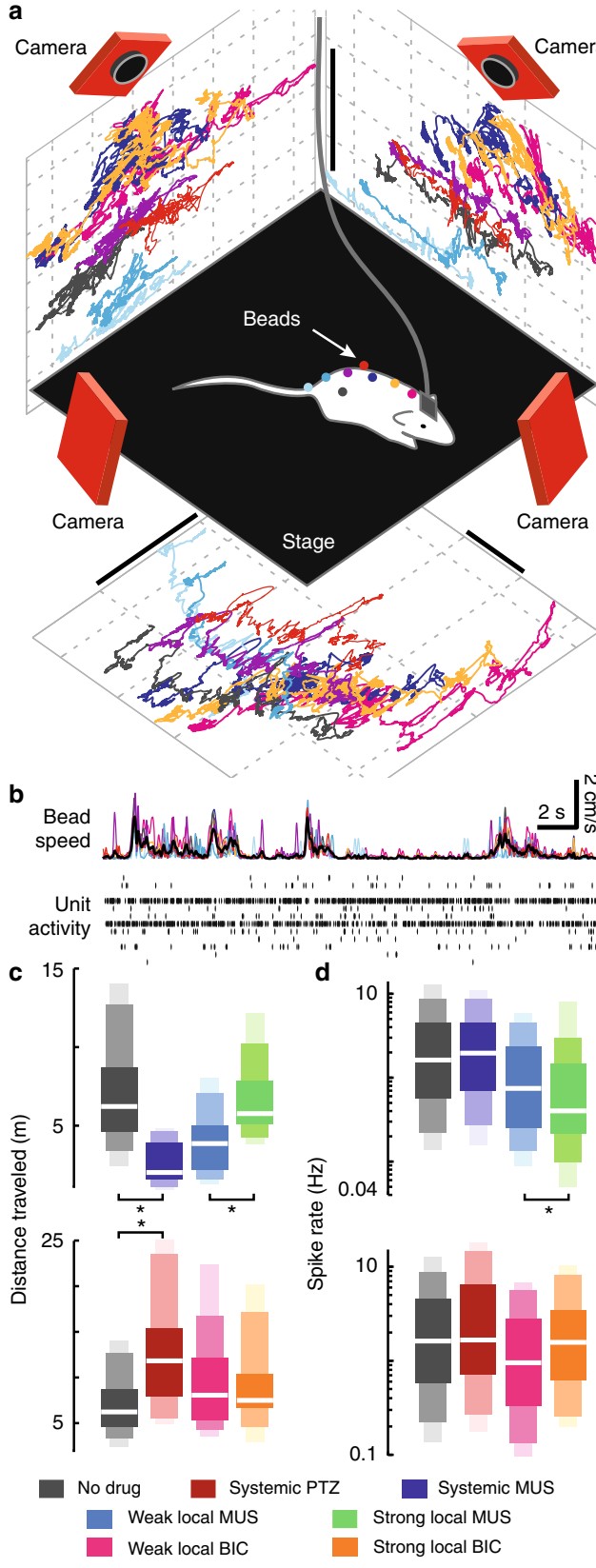

**Fig. 1** Inhibitory modulation of voluntary, unconstrained movement. **a** The positions of eight beads, attached to the body of a freely moving rat, were measured in three dimensions with submillimeter, 10 ms precision using a nine-camera motion-tracking system. Colored lines show three orthogonal projections of the bead trajectories. Scale bars indicate 5 cm. **b** We recorded the speed of all eight beads (colored lines) and analyzed their average (black) in relation to single-unit spiking activity simultaneously recorded in motor cortex. A spike raster for 12 units is shown. **c** Systemic enhancement or reduction of inhibition resulted in decreased or increased voluntary movement, respectively, compared with unaltered inhibition. Local enhancement of inhibition increased movement. **d** Spike rates were distributed lognormally and reduced by local enhancement of inhibition, but not significantly altered by other manipulations. In **c** and **d**, shaded rectangles delineate quantiles (25−75th dark, 10−90th medium, 5−95th light); white lines mark median

during the second half (Pearson's $\rho = 0.80$, $p = 0$). Moreover, the population coupling of a neuron was largely the same whether the animal was at rest or in motion ($\rho = 0.81$, $p = 0$, Fig. 2b). This observation suggests that population coupling is more strongly determined by the properties of the neuron and its connections rather than the behavioral state of the animal, consistent with previous studies in the sensory cortex[15]. In contrast with the unaltered spike rates (Fig. 1d), systemically enhanced inhibition resulted in a prominent increase in population coupling ($\rho = 0.7$, $p = 0$, Fig. 2c). Local enhancement of inhibition also increased population coupling ($\rho = 0.3$, $p = 0$, Fig. 2c). Interestingly, locally blocking inhibition also resulted in increased population coupling ($\rho = 0.2$, $p < 0.01$, Fig. 2c). These results demonstrate that population coupling is sensitive to the balance of excitation and inhibition. Moreover, the results suggest that the normal motor cortex (with unaltered inhibition) operates at a minimum in population coupling; disrupting inhibition increases population coupling whether the disruption entails enhanced or suppressed inhibition.

The most prominent change in population coupling arose for enhanced inhibition (either local or systemic.) To better understand this result, we studied a parsimonious network-level computational model (Fig. 3a, b). The model consisted of 1000 binary probabilistic integrate-and-fire neurons (80% excitatory, 20% inhibitory, more details in Methods). We measured population coupling based on a subset of 20 neurons. First, our model confirmed that increasing local inhibition decreases firing rates, as in the experiments. However, unlike our experiments, our model predicted that a purely local enhancement of inhibition should result in decreased population coupling (Fig. 3c). Considering a more holistic view of the system, our model offers a resolution of this apparent conflict between the model and the experiments. When muscimol acts globally, similar to that in our experiments with systemic manipulations of inhibition, regions that feed input to primary motor cortex (e.g, thalamus or premotor cortex) may also decrease their firing. Even for our experiments with local changes in inhibition, it is likely that our pharmacological manipulation is not strictly local, perhaps affecting nearby premotor areas of cortex. Thus, in all our experiments (both local and systemic), we should expect a decrease in the input to the neurons we recorded from. Therefore, we next used our model to test how changes in input relate to changes in population coupling. We found decreases in input can dramatically increase population coupling (Fig. 3d). Moreover, a combination of enhanced local inhibition together with decreased input (Fig. 3e), which is the case for our experiments, can accurately reproduce our experimental findings of muscimol-induced increases in population coupling (Fig. 2c).

population average), but for some neurons population coupling was negative (anticorrelated). Across units, population coupling varied over three orders of magnitude. However, it was rather stable on the timescale of one recording. Population coupling for the first half of each recording was strongly correlated with that

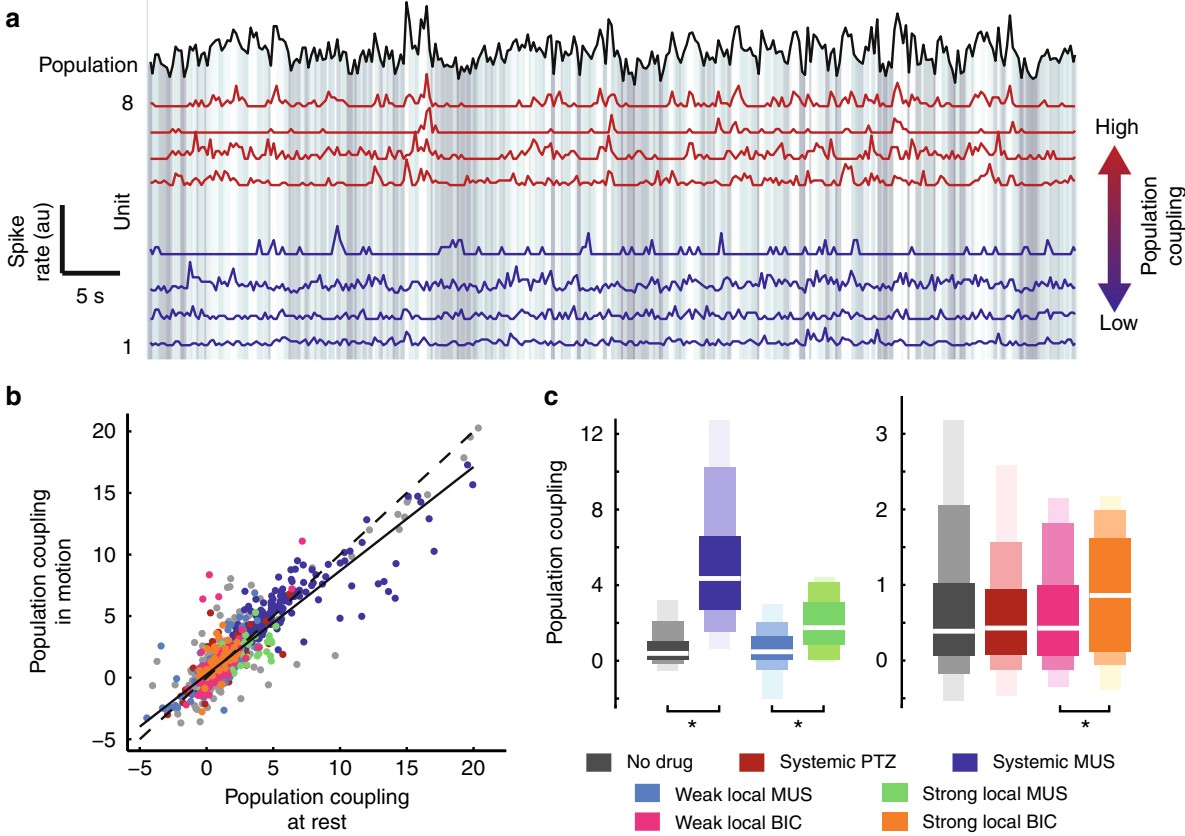

**Fig. 2** Minimal population coupling for unaltered inhibition. **a** Spike count time series for the collective population activity (black, summed over all units) and eight example single units (red—high population coupling; blue—low population coupling). Background grayscale indicates the population spike rate. **b** Population coupling during periods of rest is slightly higher, but strongly correlated with population coupling during motion. Each point represents one unit. Dashed line indicates equality. Solid line indicates best linear fit. Color indicates inhibitory condition as in **c**. **c** Enhancing inhibition results in more neurons with high population coupling, whether the manipulation was systemic or local. Locally reducing inhibition also increased population coupling. In general, unaltered inhibition resulted in minimal population coupling, averaged over units

According to its mathematical definition, population coupling could depend on the number of neurons in the recorded population. We confirmed that in our experimental results there is no correlation between population coupling and the number of units recorded (Supplementary Fig. 2). Therefore, we pooled data from multiple recordings. In our model, we kept fixed the number of examined units so that this is not an issue. We also found that population coupling was not correlated with firing rate for the rats with local inhibitory manipulations. For the systemic inhibitory manipulations, there was a weak anticorrelation between spike rate and population coupling ($\rho = -0.17$, $p < 10^{-5}$).

**Peak body coupling for low population coupling.** Next, we quantified the "body coupling" of each neuron, i.e., how strongly each single unit's firing was related to the rat's body movements. We did this in two ways, examining two different properties of each unit: movement-triggered-average spike rate (MTASR) and spike-triggered-average body speed (STABS).

First, for our analysis based on MTASR, we focused our analysis on periods of time when the rat voluntarily initiated or terminated body movement (Fig. 4a). Movement onset and cessation were defined, respectively, as the moment when the body speed of the rat exceeds or drops below the time-averaged body speed. The time-averaged body speed is close to zero, because the rats spent considerable time at rest. We found that different neurons exhibited diverse changes in firing rates related

to movement; all combinations of increased, decreased, and unchanged firing were observed for both movement onset and movement cessation (Fig. 4b), consistent with previous findings[31–33]. We defined body coupling $BC_M$ as the standard deviation (SD) of the MTASR waveform. Thus, neurons with a flat MTASR waveform (e.g., bottom row in Fig. 4b) have low $BC_M$ and neurons with strong modulation apparent in the MTASR (e.g., top two rows of Fig. 4b) have high $BC_M$. This was done separately for both movement onset events and cessation events. We found that neurons with high $BC_M$ for movement onset typically had high $BC_M$ for movement cessation as well ($\rho = 0.86$, $p = 0$, Fig. 4c). Finally, we averaged the two values for movement onset and cessation to obtain one $BC_M$ value for each neuron for comparison with population coupling (Fig. 4d). For all drug conditions, ~31% of neurons exhibited significant $BC_M$. Here, significance was judged as having higher $BC_M$ than 95% of 1000 surrogate control values obtained by shifting spike times relative to body movement times. $BC_M$ was negatively correlated with spike rate for both local ($\rho = -0.8$, $p = 0$) and global ($\rho = -0.8$, $p = 0$) inhibitory manipulations. Similar to population coupling, $BC_M$ was rather stable on the timescale of one recording. $BC_M$ for the first half of each recording was strongly correlated with that during the second half (Pearson's $\rho = 0.74$, $p = 0$).

We found that neurons with high population coupling had low $BC_M$, whereas the neurons with high $BC_M$ had low population coupling (Fig. 4d). This qualitative finding was consistent across different inhibitory manipulations. Moreover, the neurons with

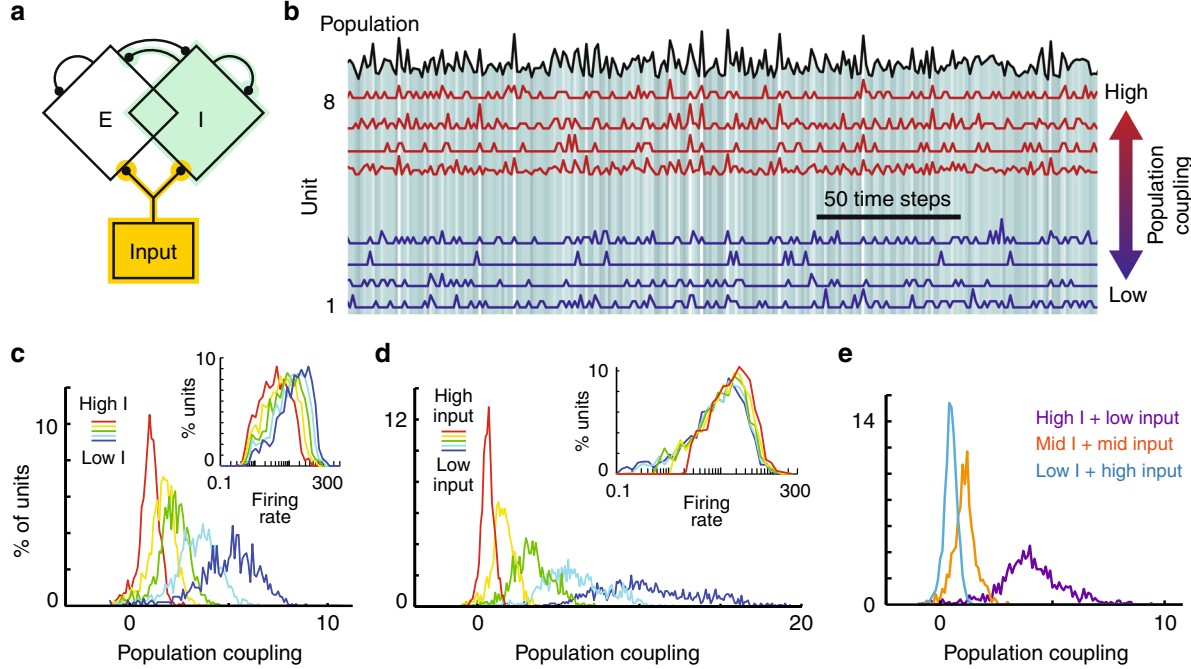

**Fig. 3** Weak external input entails strong population coupling. **a** Schematic of a model network of 1000 binary excitatory (E) and inhibitory (I) neurons driven by external input. Two components of the model change due to altering inhibition: local inhibitory input (green) and the strength of external input to the network (yellow). **b** Spike count time series from eight model neurons with high (red) and low (blue) population coupling. The black line and background grayscale indicates the population spike rate. **c** Holding external input fixed, the mean and variance of population coupling decreases as local inhibition is increased. This scenario is inconsistent with our experiments. Firing rates increase as local inhibition is decreased (inset), also inconsistent with our experiments. **d** Holding local inhibition fixed, the mean and variance of population coupling decreases as input increases. Firing rates remain relatively unchanged (inset). **e** A combination of increased local inhibition and decreased external input (purple) can match the experimentally observed increases in population coupling due to muscimol

the lowest values of population coupling also had low body coupling. Thus, there was an optimal, rather low value of population coupling with peak body coupling (Fig. 4d). The quantitative value of the optimal population coupling shifted to a higher value when inhibition was enhanced (Fig. 4d). For each condition (unaltered, enhanced, and reduced inhibition), the non-monotonic, peaked relationship between $BC_M$ and population coupling was statistically significant at a level of $p < 0.02$ (see Methods). Interestingly, for local enhancement of inhibition, all the neurons were on the left side of the peak, whereas local reduction of inhibition resulted in most neurons on the right side of the peak (Fig. 4d).

Next, we tested a different definition of body coupling based on the STABS for each unit. This definition of body coupling $BC_S$ is not biased toward the somewhat arbitrarily defined events of movement initiation and termination considered above in the $BC_M$ analysis. For each unit, we obtained the STABS waveform for the period preceding and following the spike trigger time by ± 1 s (Fig. 5a). To quantify body coupling we computed the SD of the STABS waveform. A flat STABS waveform would occur if spikes are independent of movement, resulting in a low $BC_S$ value. Consistent with the diversity of MTASR waveforms (Fig. 4b), the STABS waveforms were diverse; some exhibited sharp increases, other broad decreases, etc. (Fig. 5b). Approximately 44% of neurons exhibited significant $BC_S$ values. We found that neurons with high $BC_M$ tended to have high $BC_S$ as well; these values were correlated ($\rho = 0.7$, $p = 0$, Fig. 5c), but often quite different, suggesting that $BC_S$ and $BC_M$ provide somewhat different views of how a neuron is related to body movement. $BC_S$ was negatively correlated with spike rate for both local ($\rho = -0.5$, $p = 0$) and global ($\rho = -0.6$, $p = 0$) inhibitory

manipulations. $BC_S$ for the first half of each recording was strongly correlated with that during the second half (Pearson's $\rho = 0.62$, $p = 0$). We note that, in principle, a neuron could begin firing only during periods of sustained high body speed, which would result in a flat STABS waveform and, thus, low $BC_S$. However, in our experiments, such constant high-speed motion was extremely unusual.

We found that our primary conclusions held for this new definition of body coupling. The highest values of $BC_S$ were found for neurons with low population coupling and the neurons with strongest population coupling exhibited low $BC_S$ (Fig. 5d). For all experiments, we observed a peak in $BC_S$ at low but nonzero population coupling. The peak was statistically significant with $p < 0.02$ (see Methods).

In Figs. 4d and 5d, population coupling was calculated based on a rather coarse time resolution with the spike count computed in 250 ms time bins. Next, we sought to determine whether the peaked relationship between body coupling and population coupling also holds at finer timescales. We tested 100, 50, and 10 ms time bins. First, we found that population coupling at 250 ms resolution was highly correlated with that computed with different time resolutions (Supplementary Fig 5a, $\rho = 0.97$ for 100 ms, $\rho = 0.93$ for 50 ms, $\rho = 0.77$ for 10 ms, all $p < 10^{-10}$). More importantly, we found that for all of these temporal resolutions and for both types of body coupling, the peaked relationship remained statistically significant with $p < 0.05$, except for $BC_M$ at 10 ms and 50 ms resolution, which was not significant (Supplementary Fig. 5b).

The results shown in Figs. 4d and 5d include all recorded units pooled across different recordings and across the six rats. Considering that variability across recordings was substantial

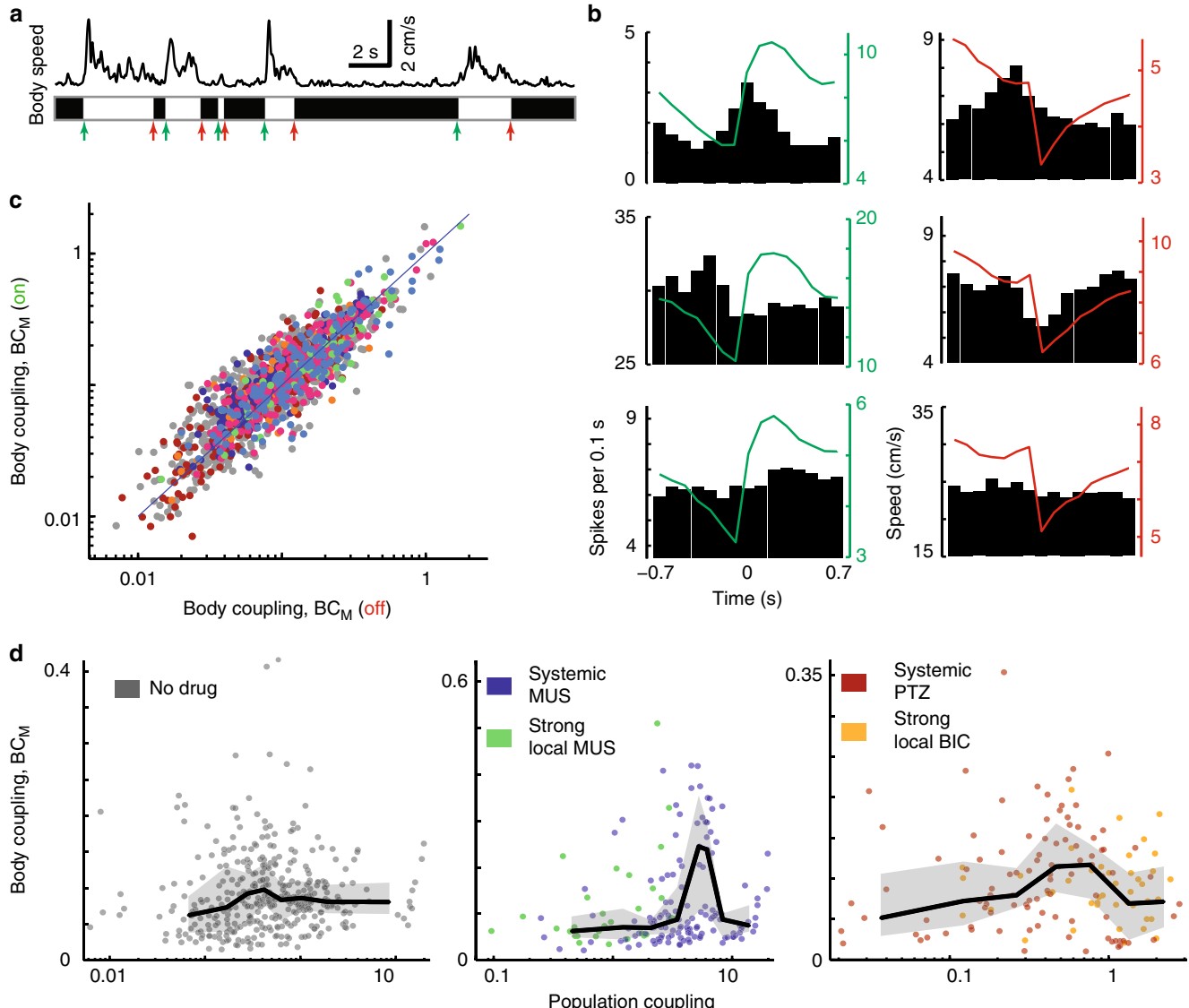

**Fig. 4** Body coupling $BC_M$ peaks at optimal weak population coupling. **a** We defined "body coupling" $BC_M$ for each unit based on how much its firing rate is modulated at body movement onset (green arrows) and cessation (red arrows). Movement onset and cessation were defined by when the average body speed increases above and drops below its mean, respectively. **b** Example movement-triggered-average spike rate histograms (black) show wide variety of relationships between spiking and movement. Movement onset (green) and movement cessation (red) were considered. Red and green lines show change in body speed averaged over the onset and cessation event trigger times. **c** Single units whose spike rate was strongly modulated by movement onset were also modulated by movement cessation. **d** We observed a peaked relationship between $BC_M$ and population coupling. Enhancing inhibition made the peaked relationship between body coupling and population coupling more prominent and shifted it toward higher population coupling. Line indicates moving average. Shaded region delineates quartiles

(Supplementary Fig. 6), this raises the questions: is our primary result—i.e., a peaked relationship between body coupling and population coupling—apparent at the single-animal and single-recording level, without pooling? First, we tested our findings within single recordings. For each single recording with at least 5 units recorded (this included 129 recordings), we fit a second-order polynomial to the body coupling vs. population coupling data (examples shown in Fig. 6a, b). We used the shape of the best fit line to judge consistency with our main results. We found that a well-above-chance number of single recordings were consistent with our main result of a peaked relationship between $BC_M$ and population coupling (77 out of 129 sessions, $p < 0.02$, see Methods). We also found that 82 out of 129 sessions were consistent with our peaked $BC_S$ vs. population coupling result ($p < 0.002$, see Methods). Next, we tested our findings for individual

animals. For both $BC_M$ (Supplementary Fig. 3) and $BC_S$ (Supplementary Fig. 4), we found that most single animals were consistent with our pooled results, showing a peak. Finally, we tested how variability across recording sessions contributed to our results. For each session, we normalized body coupling and population coupling values by the median for that session, thus reducing session-to-session variability. Our results shifted, but were qualitatively improved by this normalization (Fig. 6c, d), which suggests that within-session variability is important, and that session-to-session variability does not explain our results. For these single-session-normalized results, body coupling and population coupling exhibited a significant peaked relationship ($p < 0.001$, see Methods). Thus, we can conclude that our main findings are not simply an artifact of pooling multiple experiments. At the single animal, and even at the single-recording

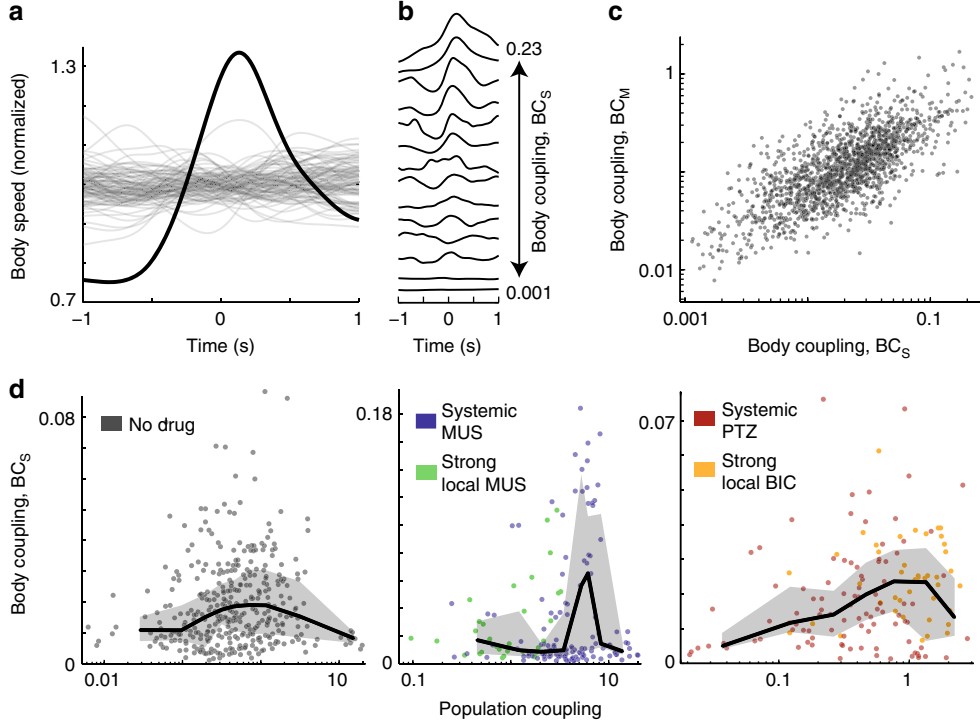

**Fig. 5** Body coupling $BC_S$ peaks at optimal weak population coupling. **a** As an alternative way to quantify body coupling, we define $BC_S$ as the SD of the spike-triggered-average body speed (STABS) waveform. The STABS waveform is normalized by its mean value. Gray lines represent 100 surrogate STABS waveforms based on temporally shifted spike times. **b** Example STABS waveforms for 14 units illustrate strong (top) to weak (bottom) body coupling. **c** Units with high $BC_M$ tended to have high $BC_S$. **d** A peaked relationship was found between $BC_S$ and population coupling. Line is a moving average. Gray shaded region delineates quartiles

level, our measurements are consistent with our main finding of a peaked relationship between body coupling and population coupling.

All of our results, up to this point, have been based on the body speed, averaged over all eight motion-tracking beads. This raises important questions. Are we averaging away important details of body motions? If we defined body coupling based on more detailed aspects of body movement, would our results still hold; would we still find a peaked relationship between body coupling and population coupling? To address these questions, we defined a third type of body coupling, called $BC_D$, which is similar to $BC_S$, but accounts for more detailed aspects of body motion. First, we analyzed the movements of the 8 tracking beads to extract 27 aspects of motion including: (1) the center-of-mass speed of the beads, (2) the angular speed of right–left turning motion, (3) the angular speed of up–down rearing motion, (4–27) three orthogonal components of bead velocity measured relative to the body center line (see Methods). Then, we computed the spike-triggered average waveform for each of these aspects of motion (Fig. 7a). To make them comparable to each other in magnitude, we z-scored them relative to 100 surrogate waveforms based on spike times with a random time shift relative to the body data. We found that different units exhibited complex relationships with these aspects of body motion. Some units were strongly related with angular turning motion, others with the center-of-mass motion, and others with relative velocity of particular beads (Fig. 7a). We define $BC_D$ for a single unit based on the aspect of movement, which has the largest "bump" in its spike-triggered average. Similar to the definition of $BC_S$, the size of a "bump" is computed as the SD (across time) of the z-scored spike-triggered average waveform. Thus, e.g., if a unit fires strongly for a slight leftward movement of the rat's head, but fires independently of the center-of-mass motion, $BC_D$ will be high, reflecting this

relationship. We found that our results held for $BC_D$; the relationship between population coupling and $BC_D$ exhibited a statistically significant peak ($p < 0.03$, Fig. 7b).

## Discussion

In conclusion, we report that during unconstrained, untrained body movement, neurons in the motor cortex are functionally segregated according to how strongly they are coupled to the population in which they are embedded. We observe a tendency for a peaked function relating body coupling to population coupling. Those neurons with extremely high or extremely low population coupling are weakly related to body movement. Those neurons with the strongest relationships to body movement have intermediate, low population coupling.

What are the implications of our findings in the context of the theoretical benefits and drawbacks of coordinated firing and population coding, as discussed above in our introduction section? The peaked nature of the relationship between body coupling and population coupling suggests that the motor cortex may make a compromise, which balances benefits and drawbacks. The neurons that are most associated with body movement—i.e., those with peak body coupling—have small but nonzero population coupling. They are weakly coordinated but not completely asynchronous. This suggests that the output population of the motor cortex might balance competing needs; trading high-capacity motor code (a benefit of weak coordination) for some robustness (a benefit of strong coordination). Similar trade-offs with optimal performance at intermediate levels of coordination have been observed in other aspects of cortical function as well[25,34–37]. However, there remains a substantial population of neurons in the motor cortex that have much higher levels of coordination among themselves, but are weakly associated with

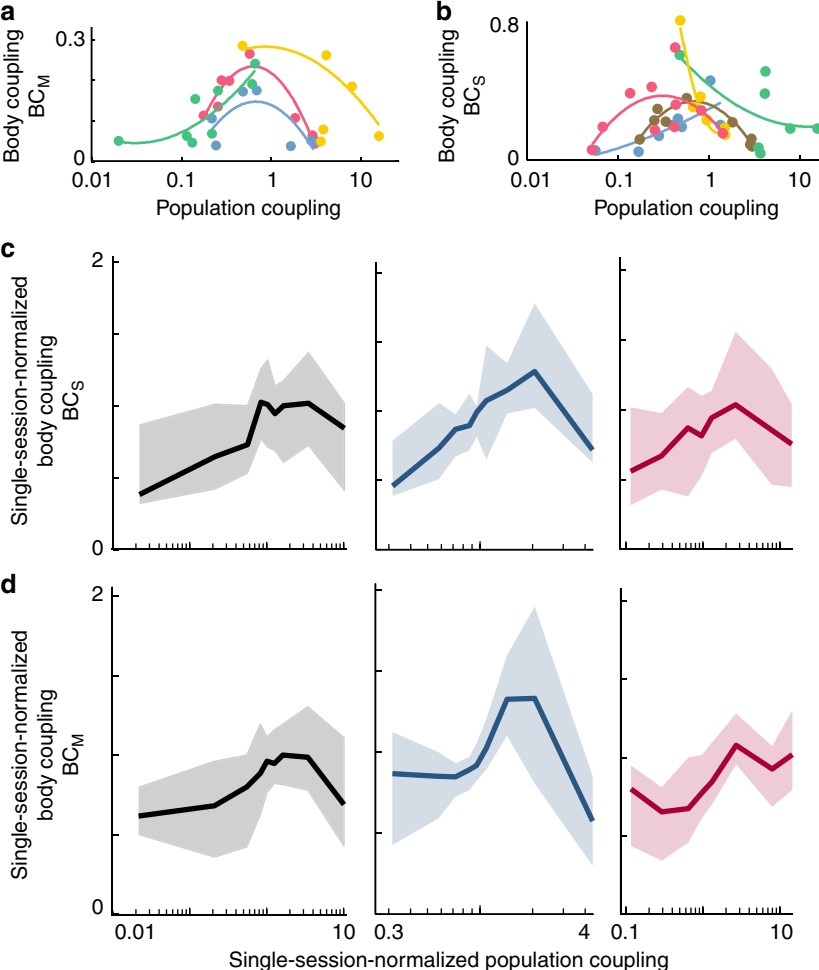

**Fig. 6** Results hold within single recording sessions. **a** Data from four recording sessions are shown (color indicates different sessions). Each point represents one unit. Each line is a quadratic fit to the corresponding points. Similar to these examples, a significant number of single sessions were consistent with the peaked relationship between $BC_M$ and population coupling found for the pooled results in Fig. 4d. Consistency with the peaked relationship was judged based on the shape of the fit line (see Methods). **b** For $BC_S$, a significant number of single session results were consistent with the pooled results in Fig. 5d. **c**, **d** The peaked relationship between body coupling and population coupling persisted after reducing session-to-session variability. Panels **c** and **d** represent $BC_S$ and $BC_M$, respectively. Each single session was normalized by its median value. Line indicates moving average; shaded region delineates quartiles. Black, blue, and red represent the unaltered, enhanced, and reduced inhibition cases, respectively

body movements. The functional role of these neurons is less clear, but could be involved with diverse internal operations of the cortex. Perhaps their higher level of coordination facilitates robust signal exchange with other cortical areas, but this remains to be tested.

It is interesting to compare our findings with the pioneering study of population coupling by Okun et al.[15] in visual cortical areas. There, it was shown that neurons in the sensory cortex with strong population coupling were strongly engaged in sensory function. More specifically, population coupling was correlated with response to visual stimuli (in V1) and visuomotor actions (in V4). Our findings, in contrast, show that neurons in the motor cortex with strong population coupling are weakly engaged by motor function. This switch from one functional principle to another between visual and motor cortical areas highlights a specific difference in potential coding strategies for input vs. output operations in cortical circuits. It would seem that output operations employ neurons with weak population coupling, whereas input operations employ neurons with strong population coupling. One implication of this point might be in optimizing brain–machine–interface systems. By choosing neurons with low population coupling (but not too

low), the control of the machine might be more successful, taking advantage of the "natural" principle we have found here. Choosing neurons with high population coupling, in contrast, might make control worse, because they are more influenced by rich, noisy internal cortical dynamics, which could complicate control signals. Future experiments would be required to test this idea.

One similarity between our results and those found in visual areas was that population coupling was largely unchanged during ongoing activity and times of functional engagement (visual stimulation[15] or, in our case, body movement). In this context, our results are consistent with the possibility that population coupling is a stable property of a neuron, perhaps governed by network anatomical structure as shown by Okun et al.[15]. However, our experiments and model demonstrate that anatomical structure is not the sole determining factor for population coupling; it can also be tuned by manipulating the balance of excitation and inhibition. Our computational model further suggests that the distribution of population coupling in a cortical network is rather sensitive to the input that network receives from external sources. Reduced input can substantially increase both the mean and variability of population coupling.

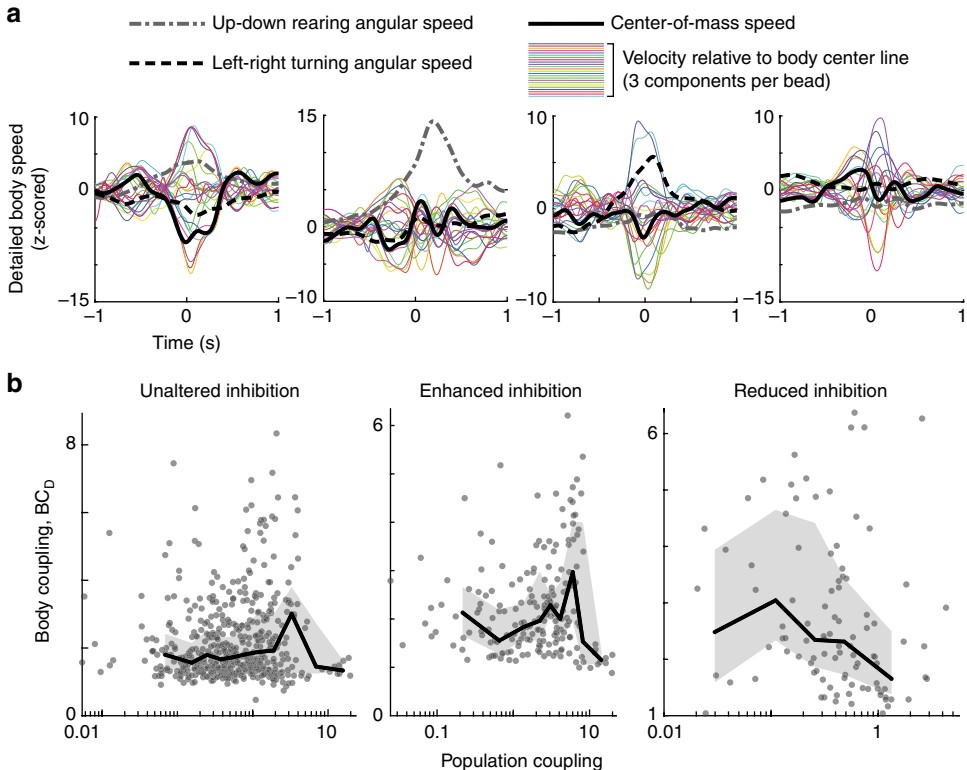

**Fig. 7** Results hold for detailed body movements. **a** Each line shows a spike-triggered average waveform for one aspect of body movement including center-of-mass speed (black), left–right rotational speed (dashed), up–down body pitch angular speed (dash dot), and three orthogonal velocity components of each bead relative to the body center line (24 colored lines). Each trace is z-scored relative to 100 surrogate waveforms based on randomly time-shifted spike times. Different panels correspond to different units. Considering all 27 aspects of movement, $BC_D$ quantifies the "bump" size for the spike-triggered waveform with the largest "bump" for each unit. **b** Body coupling $BC_D$, similar to $BC_S$ and $BC_M$, peaks for intermediate population coupling. Each gray point represents one unit. Black line is a binned average of the points. Gray shaded area delineates quartiles

Our results indicate that increased inhibition not only increases population coupling, but also changes the relationship between population coupling and body coupling. This finding may have implications for brain disorders associated with increased inhibition, such as Rett syndrome[38] and Down syndrome[39]. Our results show that when inhibition is increased, the neurons with strong body coupling shift toward higher population coupling. This means that these output neurons become more strongly influenced by the ongoing population activity within the cortex. This leaking of cortical "noise" into the output signals to the body could result in abnormal motor function. Future work may test this possibility directly in animal models of Rett syndrome and Down syndrome.

Our results demonstrate that a complete description of motor cortex must account for both the coordinated collective activity generated by some neurons (those with strong population coupling) and the asynchronous firing of other neurons (those with weak population coupling). In our study, distinguishing these differences in how single neurons relate to the population revealed fundamental differences in their functional roles. Our results suggest that neurons in the motor cortex can be categorized along a continuum. At one extreme, "internal" neurons engage strongly with each other (high population coupling), but have little to do with controlling body movement. At the other end of the continuum, "external" neurons fire relatively independently of each other (low population coupling) and issue commands that control the body. Excessive inhibition can disrupt this scenario, causing neurons that are normally external to become more strongly influenced by internal fluctuations.

## Online methods

**Animals**. All procedures were carried out in accordance with the recommendations in the Guide for the Care and Use of Laboratory Animals of the National Institutes of Health and approved by University of Arkansas Institutional Animal Care and Use Committee (protocol #14048). We studied adult male rats ($n = 6$, Rattus Norvegicus, Sprague–Dawley outbred, Harlan Laboratories, TX, USA). Given the animal-to-animal variability and complexity of the data analysis, there is no feasible way to pre-specify either an effect size or a good number of experiments. We found that three animals (~40 recordings per animal) for each condition were sufficient to obtain significant results, accounting for multiple comparisons. No randomization method was used in assigning animals to group 1 or group 2.

**Electrophysiology**. We studied two groups of rats. For group 1, microelectrode arrays (A8x4–2mm-200–200–413-CM32, Neuronexus) were chronically implanted with shank tips at a depth of 1300 μm from the pia, thus targeting most electrodes to deep cortical layers of primary motor cortex. Here we report recordings that were taken at least 2 weeks after implantation surgery. For group 2, we used a different type of microelectrode array (Buzsaki32-CM32, Neuronexus), which has electrodes that are spaced more densely in space. For both groups, the electrode arrays were oriented such that the plane of electrodes was perpendicular to the dorsal surface and parallel to the midline. The electrodes spanned 1.4 mm in the rostrocaudal direction, centered at a point 0.5 mm caudal from bregma and 2 mm lateral from midline. The probe position was chosen deliberately to sample from neurons that are associated with

a wide range of different body motions. Considering previous intracortical microstimulation studies, the region we sampled is involved in many aspects of body movement including hip flexion, trunk movements, pronation, wrist extension, elbow flexion, neck movement, and vibrissa movement[40]. The Buzsaki type probes were chosen for our second set of recordings with the goal of improving spike sorting[41]. In addition, the rats in group 2 had a microcannula included in the chronic implant for local drug delivery (26GA guide cannula, 33GA injection cannula, Plastics One, Roanoke, VA, USA). The guide cannula was implanted with its tip touching, but not penetrating the cortical surface about 500 μm from the point where the electrodes were inserted. Broadband recordings (30 min duration) of extracellular voltage fluctuations were performed with 30 kHz sample rate (Cerebus, Blackrock Microsystems). Signals were digitized by a lightweight circuit (1 cm from implant) and then transmitted via a commutator to the recording system. The wire between the rat and the commutator was spring supported, such that minimal vertical forces were applied to the rat when the rat's head was at a natural height relative to the stage, thus facilitating free movement of the rat. After filtering (250–5000 Hz band pass), potential spike waveforms were cut based on negative threshold crossing and sorted based on principal components of spike waveforms using Offline Sorter (Plexon), for group 1 rats. For group 2 rats, spike sorting was done with the Klusta software (https://github.com/kwikteam/klusta), which was developed for electrode arrays with many closely spaced recording sites, such as our Buzsaki style probes, as described previously[41].

**Motion tracking**. A nine-camera motion tracking system (Flex: V100R2, Naturalpoint) was used to track the three-dimensional positions of eight reflective beads (MCP1125, Naturalpoint, 3 mm diameter), temporarily adhered along the rat's neck, back, rear hips, and base of tail. The camera frame rate was 100 Hz and the system tracked the bead positions with submillimeter resolution. The rats were placed on a 30 cm × 30 cm square stage inside of a dark box and allowed to move freely, without constraint or trained task, during each 30 min recording. Each rat underwent three acclimatization sessions before recording to avoid excessive stress. Each acclimatization session was 15–20 min in duration. The position of each bead was first smoothed by low-pass filtering (cutoff at 5 Hz) and then differentiated to compute the speed of each bead.

**Pharmacology**. For group 1 rats, 1 h before every recording the rat was given an IP injection of either sterile saline (sham condition), saline + muscimol (2 mg/kg), or saline + PTZ (30 mg/kg). Muscimol is a GABA$_A$ agonist[42] and PTZ is a known convulsant that binds to GABA$_A$ receptors[43] and increases neuronal excitability by affecting calcium channels[44]. One sham recording (first) and one drugged recording (second) were performed on each recording day for each rat. The time between consecutive recordings on one day for one rat was 1.5 h. One or two day breaks between recording days were given.

For group 2 rats, the small volumes (1–2 μL, 0.2 μL/min for 5 or 10 min) of drug (muscimol or bicuculline methiodide) dissolved in sterile saline or just saline (sham condition) was injected through the microcannula. The injection was done using a syringe pump (Bioanalytical Systems, Inc., IN, USA). Bicuculline is a GABA$_A$ antagonist[45]. Multiple concentrations were tested for both muscimol and bicuculline including 20, 40, 80, 160, 320, 640, and 1280 μM. We found that many of the effects of the local drug manipulations were quite small for the lower drug concentrations. To obtain a clearer view of drug effects, we divided the experiments into two groups for some of our results. The "strong" group included concentrations 320, 640, and 1280 μM; the "weak" group included 20, 40, 80, and 160 μM.

**Data analysis**. Population coupling of neuron $i$ was calculated as defined in prior studies[15],

$$C_{\text{pop},i} = \frac{1}{N_i} \sum_{t=1}^{T} f_i(t) P_i(t), \tag{1}$$

where $f_i(t)$ is the spike count of neuron $i$ in time bin $t$, $N_i$ is the total number of spikes for neuron $i$ over the entire duration of the recording, and the population spike count time series is given by

$$P_i(t) = \sum_{j \neq i} \left( f_j(t) - \mu_j \right), \tag{2}$$

where $\mu_j$ is the mean spike count for neuron $j$. We chose time bins of duration 0.25 s to approximately match the minimal timescales of body movement, but our results were robust to changes in the choice of time bin (results for time bins with 100, 50, and 10 ms shown in Supplementary Fig. 5). Clearly population coupling will be poorly estimated if the total number of recorded neurons is small. Thus, we excluded from analysis all recordings with fewer than 5 units recorded.

We studied body coupling defined in three different ways. One type of body coupling BC$_M$ was defined using the MTASR waveform. Two types of MTASR waveforms were constructed. First, movement onset times were used for triggers. Second, movement cessation times were used for triggers. Spike rate was calculated in a period ± 1 s around the movement trigger times with 10 ms time resolution. The MTASR waveform was low-pass filtered at 1.5 Hz and then normalized by its mean. Then, the SDs of the onset MTASR and the cessation MTASR were computed. BC$_M$ was defined as the mean of the onset and cessation values of SD.

The second type of body coupling, BC$_S$, was defined for each unit as the SD of the STABS waveform. The STABS waveform was constructed for a period ± 1 s around the spike times of the unit with 10 ms time resolution. The waveform was low-pass filtered at 1.5 Hz and then normalized by its mean. Finally, BC$_S$ was computed as the SD of this filtered, normalized STABS waveform.

The third type of body coupling, BC$_D$, was defined to account for more detailed aspects of body movement. BC$_D$ was calculated based on multiple spike-triggered-average detailed body movement (STADBM) waveforms. For each unit, we constructed a STADBM for the 27 aspects of body movement described in the main text. The steps for extracting the 27 aspects of body motion are as follows. First, we calculated the center-of-mass position at every time point (equal mass weighting of each of the eight beads). This was used in two ways as follows: (1) to compute center-of-mass speed and (2) to help define the body vector. The body vector was defined as the best fit line (in three dimensions) to three points including the center of mass, the head tracking bead, and the tracking bead near the base of the neck. The end of the fit line closest to the head defined the pointing direction of the body vector. Two angles were then computed based on the body vector at each time point. The left–right turning angle was defined as the angle between a fixed horizontal line and the projection of the body vector onto a horizontal plane. The up–down rearing angle was defined as the angle between the body vector and its projection onto a horizontal plane. Finally, the three orthogonal components of velocity of each tracking bead were computed relative to the body vector. The directions of these components were (1) along the rostrocaudal body axis, (2) the horizontal right–left axis, relative to the body, and (3) up–down axis relative to horizontal. This relative motion captured changes in posture, body curvature, and other small motions relative to the body axis.

Both population coupling and body coupling are prone to poor estimates for neurons with very low spike rates. To avoid such low sampling errors, we excluded from our analysis neurons with

spike rates $< 0.5$ Hz. We also tried different firing rate thresholds (0.5, 0.2, 0.1 Hz) and found that our results did not change much.

In several places in the manuscript, we examine the relationship between two quantities with a Spearman's correlation coefficient and a corresponding $p$-value. The $p$-value represents the probability of the null hypothesis that the two quantities are uncorrelated. The $p$-value is calculated based on many random shuffles of the data ordering.

To assess the statistical significance of a non-monotonic peaked relationship between body coupling ($y$) and population coupling ($x$), we performed the following several steps. First, considering a peaked relationship between $y$ and $x$, then we call $x^\star$ the particular value of $x$, which is closest to the peak in $y$. To identify $x^\star$, we tested every measured value of $x$, except for a small number (at least 10) of points at the extremes of the $x$ range. For each possible choice of $x^\star$, we computed a Spearman's correlation coefficient $\rho_{\text{left}}$ for the points to the left of the peak (those with $x < x^\star$) and another correlation coefficient $\rho_{\text{right}}$ for the points to the right of the peak (those with $x > x^\star$). A peaked relationship should give $\rho_{\text{left}} > 0$ and $\rho_{\text{right}} < 0$. We excluded the choices of putative $x^\star$ that did not meet this criterion. For each putative $x^\star$, we then defined the peakiness $P$ to be the smaller of $\rho_{\text{left}}$ and $-\rho_{\text{right}}$ (i.e., high $P$ requires both sides of the peak to have strong correlations). $P = 1$ for a noiseless, perfect peak, and near zero for uncorrelated data. Then we chose $x^\star$ to be the value of $x$ that maximizes $P$. Then we repeated this process for 1000 surrogate shuffled datasets for which population coupling values were randomized relative to the body coupling values. For each shuffled dataset, we computed the best $x^\star$ and its corresponding peakiness $P$ (in cases where there was no valid $x^\star$, we set $P$ to 0). The $p$-value reported in the main text is the probability of finding a surrogate shuffled dataset with a greater $P$ than our actual measured $P$.

For statistical assessment of whether single recording sessions were consistent with the peaked relationship between body coupling and population coupling, we did the following. First, we fit a second-order polynomial to the body coupling vs. population coupling points for the recording. Only recordings with at least 5 units were considered. The recording was deemed consistent with our main finding if either of two criteria were met for the best fit polynomial. The first criterion was that the fit have a negative coefficient for the quadratic term, i.e., downward curvature (as all peaks have), and that the maximum of the fit be within 10% from the peak found for the pooled data. As mentioned above, the peak for the pooled data was the value of $x^\star$ with the highest $P$. In some cases where the first criterion was not met, most of the units had population coupling values that were not spanning the peak found for the pooled data, falling mostly to the left or right of the peak instead (examples shown in Fig. 6a, b). In this case, we tested a second criterion based on the slope of the fit. If most of the units were left of the peak and the slope was positive, the recording was deemed consistent with our main result. If most of the units were right of the peak and the slope was negative, the recording was deemed consistent with our main result. Here, to be more precise, "most of the units" means more than half of the range of population coupling spanned by the measured units. The slope was averaged over the range of the fit that was either to the right or left of the peak, according to where most of the units were. Finally, after counting the number of single recordings that were consistent with our main findings, we assessed the statistical significance of this count by repeating the whole process with randomized data (body coupling values shuffled across units and recordings, without changing population coupling). The randomization resulted in a new peak location for the pooled data and a randomized set of points within each recording for fitting the second-order polynomial.

Investigators were not blinded to different groups and conditions during data analysis.

**Computational model.** The model consisted of $N = 1000$ binary neurons. The state $s_i(t)$ of neuron $i$ at time $t$ is either 0 (not spiking) or 1 (spiking), which is determined according to the following equation.

$$s_i(t) = \begin{cases} 1 \text{ with probability } p_i(t) \\ 0 \text{ otherwise} \end{cases} \quad (3)$$

$$p_i(t) = \sigma\left(\left[\eta + \sum_{j=1}^{N} W_{ij} s_j(t-1)\right] r_i(t)\right) \quad (4)$$

where $\sigma(x)$ constrains $x$ to be between 0 and 1, $\sigma(x) = 0$ for $x \leq 0$, $\sigma(x) = 1$ for $x \geq 1$, and $\sigma(x) = x$ for $0 < x < 1$. The sum represents input from other neurons, which fired at time $t-1$ and external input to the network is represented by the constant $\eta$. Firing probability is reduced by the activity-dependent factor

$$r_i(t) = \left(1 + \chi + \alpha\left[\sum_{\tau=t-T_r}^{t-1} s_i(\tau)\right]\right)^{-1} \quad (5)$$

where the timescale of history dependence is $T_r = 100$ time steps and the magnitude $\alpha = 0.1$. Local inhibition is modeled by the variable $\chi$ ($\chi > 0$ entails enhanced inhibition, $\chi < 0$ entails reduced inhibition). In Fig. 3c, $\eta$ was held fixed at $8 \times 10^{-4}$ and we tested five values of $\chi = 0.2, 0.1, 0, -0.1$, and $-0.2$. In Fig. 3d, $\chi$ was held fixed at 0 and we tested five values of $\eta = 2 \times 10^{-4}, 4 \times 10^{-4}, 8 \times 10^{-4}, 1.6 \times 10^{-3}$, and $3.2 \times 10^{-3}$. In Fig. 3e, $\eta = 4 \times 10^{-4}, 2 \times 10^{-3}, 5 \times 10^{-3}$, and $\chi = 0.1, 0$, and $-0.1$ were used respectively for the three cases: high I + low input, mid I + mid input, and low I + high input.

The model was run for 50,000 time steps. Spike count time series were constructed using time bins of duration 50 time steps. Population coupling was computed based on groups of 20 neurons (this mimics the fact that we experimentally measured from a small subset of neurons). We obtained 1000 population coupling values, one for each neuron, based on 50 non-overlapping subsets of 20.

The $N \times N$ matrix $W$ models the network structure and synapse weights. $W$ was constructed in four steps. First, an $N \times N$ matrix of numbers was drawn from a lognormal distribution with mean 0.5 and variance 1, consistent with experimental observations[46,47]. Second, inhibitory neurons were designated by multiplying 20% of the columns of $W$ by $-1$. Third, some of the inputs for each neuron were set to zero (i.e., disconnected), such that the number of inputs (in-degree) were distributed lognormally across neurons (mean in-degree was 20, variance was 500). Such a long-tailed distribution of in-degree was important to better match experimentally observed firing rate distributions. Finally, the entire matrix was divided by a constant such that the largest eigenvalue of the matrix was 1, which ensures that the network dynamics are stable (neither growing nor decaying in time, on average), as studied in previous work[48,49].

**Data availability**
The neural and body movement data that support the findings of this study are freely available for download from Figshare (https://doi.org/10.6084/m9.figshare.7562192).

**Code availability**
The Matlab code we developed for testing the statistical significance of a peak in noisy data is freely available for download from Figshare (https://doi.org/10.6084/m9.figshare.7562192).

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

## Acknowledgements

W.L.S., S.H.G., and P.K. were supported by grant FQXi-RFP3–1343 from Foundational Questions Institute. W.L.S., S.H.G., and J.L. were supported by grants from Arkansas Biosciences Institute.

## Author contributions

P.K., S.H.G., and J.L. performed the experiments. W.L.S., P.K., and S.H.G. conceived the study. W.L.S., P.K., S.H.G., and J.L. wrote the paper. W.L.S. and L.F. performed computational modeling. W.L.S., P.K., and J.L. performed the data analysis.

## Additional information

**Competing interests:** The authors declare no competing interests.

