## [Peer Review File · Nature Communications]

Reviewers' comments:

Reviewer #1 (Remarks to the Author):

Here, Dr. Shew and group examine an interesting and timely question on how the couplings between neurons in the mouse brain relate to movement. While extensive prior work has shown that population coupling is functionally important and that it likely plays a critical role in behavior, the role that proposed 'strong' and 'weak' couplings play in this process has been less clear. The present study represents an important effort to understand this process and involves the simultaneous recordings of neural populations in the motor cortex of mice as well as the detailed recording of their body/limb positions during naturalistic unconstrained movement. The authors then use some advanced population modeling techniques to evaluate the relation between population response and movement, and find that strong couplings within the population were not involved in body movement whereas weakly coupled neurons are informative of 'neuron-body' information transmission.

Overall, the paper tests an interesting question with potentially important implications to our understanding of how movement is encoded and mediated within the brain. That being said, there are a number of major issues that would need to be first addressed including one or two significant concerns that could potentially limit interpretation of the findings.

1. It is not clear to me what specific movements are being encoded. For example, there do not appear to be any specific distinctions made between movement velocity, onset, type of movement (e.g., forelimb/hindlimb) and how they all correlate with changes in neural response.
2. Changes in movement inherently lead to concerted changes in firing rate across the population. While the authors use a modeling approach to examine for shared influences on the population, it still does not account for the fact that increase in firing rate across the population increases the strength of population couplings or that it may disproportionately increase 'strong' compared to 'weak' couplings.
3. It is difficult to interpret the results from the GABA antagonist. Specifically, a systemic GABA antagonist could influence movement through direct spinal pathways and therefore any relation between cortical activity and movement response would be confounded. Moreover, while the authors conclude that their model suggests that muscimol reduces thalamic firing and therefore input to the cortex, it doesn't take into account the possibility of other bottom-up or top-down influences.
4. It is indicated that 732 units were recorded from. However, it is not clear how many electrode arrays were these obtained from and over how many repetitions. In other words, how many 'unique' neurons were obtained?
5. Why were only 4 states evaluated in relation to movement? Also, it seems that the 0.5 second intervals used in the study are not really evaluating for spike-correlations but rather firing rate correlations. On a similar line, statistically, it is not clear how you accounted for the multiple comparisons made between all the possible body states and 4 unit states. I also find a similar issue with the STABs procedure.
6. The interpretation and conclusions of the study are not clearly stated. Even after reading the paper, it is not full apparent to me what a 'neuron-to-body information transmission' is. All you are seeing is really a difference in correlation between coupling and movement, but there is no time-causal analysis and no correspondence between specific body movement and neural activity to be considered as informative.

Reviewer #2 (Remarks to the Author):

This manuscript aims to determine the coupling between neurons and gross body movements in rodents spontaneously moving around a platform. Overall, the manuscript uses an interesting analytical framework and presents a potentially interesting finding. However, I am concerned that there are several major flaws in their approach. Primarily, it is really unclear if they are actually recording from a task-relevant region of motor cortex. For example, Figure 1b shows a surprisingly quiet pattern of activation during movements?? Similarly, there are pauses in activity during increased body speed? Obviously, the manuscript's conclusion would be very different if they were recording from a non-task modulated region. The use of systemic methods for modulating inhibition also greatly diminishes enthusiasm; it makes it very difficult to interpret the results.

MAJOR

- 1) Very little is stated in the methods about the details of the "motor cortex" recording (i.e. coordinates, general body region targeted). The task seems to be a measure of gross movement (i.e. body speed). I imagine that the rat has to walk/run to achieve this. If the recordings were in traditional forelimb or hindlimb M1, I would expect gait cycle modulation. Demonstration of gait cycle modulation or other task parameters seems very important. At the level of resolution shown, I would also imagine dramatic increases in firing rate during bursts of movement.
- 2) Did the authors exclude periods of sleep or extended immobility? This strain is also well known to demonstrate absence seizures. Did the manipulations change that aspect?
- 3) It is surprising to me that rest and movement show no difference in firing rate or coupling? What is authors explanation for this?
- 4) No spike waveforms or ISIs are shown. It is difficult to imagine that all recorded units were true single units? What was the specific criteria as I believe monotrodes were used. Multiunit recordings would of course alter the interpretation.
- 5) How much acclimatization was performed? In a well-learned environment, was any motivation given to explore/walk?
- 6) Use of a 100 surrogates seems low for the shuffling procedure.

MINOR

- 1) Provide justification for simply averaging all the markers?
- 2) Use of the 0.25 ms bin is not sufficiently explained/justified.
- 3) Please indicate why use of an untrained task is important.

Reviewer #3 (Remarks to the Author):

The authors investigated relationship between population coupling and movement coupling in the motor cortex neurons. Their main claim is that neurons with strong population coupling are weakly coupled to body movement and conversely the ones with strong body coupling are weakly coupled to population. The claim is further supported by manipulating neuron-coupling using drugs. The question about how neuron-to-neuron coupling is related to encoding of external variables is an interesting topic and drawing a lot of attention in studies of sensory coding. Here, they aim at extending this knowledge to the motor cortex. The study is novel and data are clearly presented. Yet there are major concerns regarding the main claim of the study. Unless these concerns are addressed, the main claim is not strongly supported by the data.

1. When the authors use different ways to calculate body-coupling, the results are different in a critical way. In one condition, neuron-coupling and body-coupling are anti-correlated. In the other condition, two variables are still related significantly, but in a different way. It looks more like inverted U (or V)

shape. The authors claimed that neurons with strong body coupling are weakly coupled to population, but when looking at Figure 6e, it seems more accurate to say they showed intermediate level of population coupling. Furthermore, the effect of enhanced inhibition seems to have an opposite effect on body-coupling measured in different ways. It is critical to understand the difference of these two measures.

2. Related to the previous point, I did not understand why the authors use "standard deviation" of spike-triggered-average body speed (STABS) as a measure for body-coupling. Why not use mean (, or absolute value of mean should be more appropriate). Please explain. Please also describe time points relative to the spike used to calculate standard deviation. At time 0?

3. Anti-correlation (or distance correlation in Fig 6) between population-coupling vs. body-coupling can come from two different sources. One is across-session variation of population-coupling and body-coupling. The same neuron might show different degrees of population-coupling and body-coupling across sessions and they may co-vary. The other is across-neuron variation of population-coupling and body-coupling. The latter is what authors are interested in. Thus the contamination of the first component has to be factored out. One way is to calculate correlation between population-coupling vs. body-coupling for each session and present a histogram of this correlation. Another way is to normalize population-coupling and body-coupling for each session (standardize for each sessions, for example) before putting them together.

4. Authors only use neurons that pass a certain criteria for body-coupling. Why not use all the neurons? Neurons with weak body-coupling are also part of data that can be used to assess relationship between body-coupling and neuron-coupling. What is the justification for using only a subset of neurons?

5. Even if it is somehow justified to select a subset of neurons, choice of criteria is arbitrary. It is nice to see in Figure S2 that the main claim is insensitive to the choice of criteria. But they are not showing within condition statistics for Figure S2, which is more important. Again for figure 6, insensitivity to the choice of criteria should be presented.

Minor concerns

Please cite literature claiming that Pentylentetrazol is a GABAA antagonist.

Please describe a bit more about distance correlation coefficient.

The motor cortex is a large area. Please describe the coordinates of recording.

Please describe the number of sessions for each drug condition.

Reviewer #4 (Remarks to the Author):

The authors of this paper investigate two properties of neurons, and their interaction. First, they categorize the degree to which neurons are coupled with the population, by finding the correlation between individual units' firing patterns and the firing patterns created by summing the population of the remaining units. Second, they categorize the degree to which units are coupled to the body, by finding the correspondence between individual units' firing rates and the rats' velocity (specifically, whether movements are ending or initiating). They measured these two metrics (population coding and body coding) under three different possible intraperitoneal injections: saline (sham) injection, muscimol (GABA agonist) injection, or pentylentetrazol (GABA antagonist) injection. They report that

increasing global inhibition increases population coupling and decreases body coupling. Decreasing global inhibition has the opposite effect, decreasing population coupling and increasing body coupling. They also report that during the control (saline injection) condition, individual units which are more closely linked to the population are less closely linked to the body movements, and vice-versa. They include modeling results which replicate the changes in population coupling under the different pharmacological manipulation states, under the assumption that changes in global inhibition affect both the local inhibitory units of the circuit and the overall amount of input drive to the area.

I have a number of concerns about this manuscript, which I have enumerated below. I believe the result described in this paper is potentially of interest, and could serve to inform other researchers who are attempting to link circuit properties of individual units to their role in behavior. However, I cannot recommend the paper for publication before these concerns are addressed. As written, I am insufficiently confident in the consistency and accuracy of the reported results. It is my hope that my questions and suggestions will help the authors to demonstrate their findings more rigorously and convincingly, and to better explain their significance.

1. The authors mention proposed theories as to why having units which fire in a more "coordinated" fashion might be advantageous in some circumstances (such as needing to transmit information with low noise), whereas having neurons which are more "independent" might be advantageous in others (computational efficiency). For example, in the abstract, they write: "Commands to the body might be more robustly conveyed by a strongly coupled population, while a motor code with greater information capacity could be implemented by neurons that fire more independently" (lines 9-12). However, their results seem to go against the direction of these theories. They report that units which are more related to the body are less related to other units. This is potentially interesting, but I would like to hear the authors discuss possible reasons behind these oppositions. Without any context for these results, their significance remains unclear.

2. This study involves three animals with chronically implanted array recording. This could mean that the number of units recorded is substantially smaller than reported: one cannot assume that each unit recorded on a given day was not already recorded on a previous day. At the same time, it is also impossible to assume that the units are stable throughout day-to-day recordings, which presents a significant quandary, as discussed by the authors in lines 206-207. Given the reported recording quantities as shown in figure S1, this indicates that the minimum "n" of units recorded is closer to 60 (maximum of 20 units recorded per day, for 3 animals). Now, it is likely that the real number is higher than 60, but also almost certainly the case that it is less than the reported 732. I have a couple of suggestions for ways to address these concerns that go beyond simply noting them in the discussion section:

- Analyze datasets separately, rather than pooling all units together. Given how few animals are included in the study, I would at the very least want to see the primary analyses (e.g. figure 5) broken down by animal, to ensure that the results are consistent across animals.

- Ideally, I would like a report of how many (if any) individual recording days feature the reported results (anti-correlation of "body coupling" and "population coupling"). This will help to ensure that a few repeatedly recorded units are not skewing the results by being included multiple times in the analysis, by requiring that the analysis be consistent within a single days' recording. Within a recording, units are presumably not repeated, and therefore will contribute only once to each recording's analysis.

3. In the legend for figure 5 (lines 440-442), the authors report "Among those neurons with significant body coupling (colored), high population coupling was found only for neurons with low body coupling." This has led me to be concerned that the authors' method of assessing the anti-correlation between body coupling and population coupling only took into account units with significant body coupling. Is

this the case? If so, they should change their analysis to take into account all units: presumably those with non-significant coupling should be those with, essentially, the "lowest" coupling values, and thus these units, too, are important for the analysis. Are units with no significant body coupling more likely to have large population coupling and vice-versa?

4. How stable are the metrics of population coupling and body coupling? For population coupling, the authors report that there is not a significant change between the coupling during movement or at rest. However, there seems to be little to no correspondence between the population coupling values (and the body coupling values) recorded from units on the same electrode and different days. This could be explained either by different units being recorded on the same electrode, or by drift in these values over time. This has led me to be concerned about how stable these metrics are within a recording session. If you cut the session into a first half and a second half, are the population coupling values and the body coupling values similar? This could also be something which could be assessed via the model (for population coupling): do the model population coupling values remain stable over time? Furthermore, the model is a good way to assess how likely the population coupling as measured by a subset of the population matches the population coupling using the whole population. If you measure population coupling with the entire 1000-unit population, compared with the 20-unit subpopulations, how correlated are these measurements?

5. In how much agreement are the STABS metric and the mutual information metric of body coupling? Are units which tend to be body coupled using one metric also body coupled using the other metric?

6. The authors use different thresholds to assess significance for different analyses. For example, the mutual information between the unit firing rate and the "starting/stopping movement" times had to exceed 90% of the shuffled values in order to be considered significant, whereas the STABS metric only needed to exceed 60% of the shuffled values. Furthermore, population coupling is reported in degree but not in significance, even though it seems that a "shuffled spike time" analysis similar to that for STABS would also function to determine the significance of the population coupling. I request that they always note why a particular threshold was chosen.

7. The pharmacological conditions were analyzed together when comparing population coupling to body coupling using mutual information, but analyzed separately when using STABS. Either the two metrics (STABS and mutual information) should be analyzed as similarly as possible, or a good justification should be provided. In addition, the primary result (anti-correlation of body coupling and population coupling) changes depending on the metric of body coupling which is used. Either there is a simple anti-correlation between population coupling and body coupling, or there is some "peak" (which also shifts depending on the pharmacological condition, and presumably disappears in the "reduced inhibition" conditions, given that the authors did not show this condition's results). This lack of correspondence is concerning, and I would like the authors to give more thought as to why it might have occurred, and what it means for their results.

Minor questions/comments:

- Sham and injection experiments were always done on the same days (and presumably in the same order). Is there an effect of the fact that the experimental conditions were always (presumably) the second experimental session of the day? Are the same units recorded on sham and non-sham experiments on the same day?

- In figure 4c, are these six different units, or three units whose "increased speed" and "decreased speed" tuning are both shown?

- Why is figure 5b shown in logarithmic scale, while figure 5a is shown in linear scale?

- In line 128, "can accounts" should be "can account."

- Because the STABS metric is assessed via standard deviation, it is preferentially searching for units with a strong correlation with either increases in body speed or decreases in body speed. For example, the STABS metric would not flag a unit as "body coupled" if the unit always fired when the rat was running at a high speed - The STABS metric would show a flat line that happened to be a higher speed than the shuffled trials, which would be normalized away. The authors comment that this metric "is not biased toward the somewhat arbitrarily defined events of movement initiation and termination considered above" (159-161). They should make clear the fact that STABS is also primarily searching for units which tend to fire around speeding up or slowing down events.

Authors' reply to reviewers

We thank the reviewers for their positive comments and enthusiasm for our research questions and initial results. We appreciate the reviewers' careful first read of our manuscript and their constructive suggestions and comments, which we have used to substantially improve our paper. Our most notable improvements are the following.

- New experiments with local manipulation of inhibition, better electrodes for spike sorting, and a better spike sorting algorithm. These new experiments help clarify our interpretations and have more than doubled the amount of data we present now compared to our original version.
- New data analysis methods which demonstrate that our findings are robust on a single-animal level and on a single-experiment level and with fewer arbitrary thresholds for data exclusion and new analyses.

Most importantly, our new data support our initial claims even more strongly than before and revealed a few new findings as well. Below, we provide more detailed descriptions of how these and other efforts address the concerns raised by the reviewers. Our replies are in Arial font, while the reviewers' comments are in Palatino font.

Reviewer #1 (Remarks to the Author):

Here, Dr. Shew and group examine an interesting and timely question on how the couplings between neurons in the mouse brain relate to movement. While extensive prior work has shown that population coupling is functionally important and that it likely plays a critical role in behavior, the role that proposed 'strong' and 'weak' couplings play in this process has been less clear. The present study represents an important effort to understand this process and involves the simultaneous recordings of neural populations in the motor cortex of mice as well as the detailed recording of their body/limb positions during naturalistic unconstrained movement. The authors then use some advanced population modeling techniques to evaluate the relation between population response and movement, and find that strong couplings within the population were not involved in body movement whereas weakly coupled neurons are informative of 'neuron-body' information transmission.

Overall, the paper tests an interesting question with potentially important implications to our understanding of how movement is encoded and mediated within the brain. That being said, there are a number of major issues that would need to be first addressed including one or two significant concerns that could potentially limit interpretation of the findings.

1. It is not clear to me what specific movements are being encoded. For example, there do not appear to be any specific distinctions made between movement velocity, onset, type of

movement (e.g., forelimb/hindlimb) and how they all correlate with changes in neural response.

The reviewer has correctly pointed out that our primary goal was not to study a coding strategy for any particular specific task or muscle group. Indeed, our goals were to first identify neurons that had any association with body movement and, second, to examine how these movement-related neurons differ from those neurons that have no association (or a very weak association) to body movement. To meet our goals, the spatial position and size of our electrode arrays was chosen to span multiple parts of primary motor cortex including those associated with neck, trunk, hips, wrists, whisker, and other movements. Again, the goal was to determine, for each unit, whether and how much it was engaged by any body movement, not a particular type of movement.

We acknowledge that our goal differs from the more common goal which seeks coding properties of the neurons that are engaged during a specific task. In our opinion, this traditional approach is important, but our approach is also important for two reasons. First, untrained, spontaneous movements can involve different mechanisms than trained stereotyped tasks. For instance, certain brain disorders, like autism, are associated with abnormal repetitive spontaneous non-task movements, but can have normal performance on tasks. Second, it is also interesting to examine task-independent neurons. Motor systems research often ignores the role of these other neurons in motor cortex that apparently have little to do with movement. We find that these movement-unrelated neurons have an intrinsically different relationship to the cortical network – they have stronger population coupling. Thus, although our work does not address specific coding algorithms for specific movements, it nonetheless reveals important new information on motor cortex.

2. Changes in movement inherently lead to concerted changes in firing rate across the population. While the authors use a modeling approach to examine for shared influences on the population, it still does not account for the fact that increase in firing rate across the population increases the strength of population couplings or that it may disproportionately increase ‘strong’ compared to ‘weak’ couplings.

We thank the reviewer for highlighting the role that firing rates could play in our findings. To address this possibility, we have now directly calculated the correlation coefficient between firing rates and the quantities of interest in our study and reported them in our Results section. The reviewer is correct that firing rate is strongly related with some of these quantities. In summary, we found that firing rate was...

- Strongly anticorrelated with body coupling BC_S based on STABS analysis for both global ($\rho=-0.6$, $p<10^{-80}$) and local ($\rho=-0.5$, $p<10^{-60}$) inhibitory manipulations.

- Strongly anticorrelated with body coupling BC_M based on MTASR analysis for both global ($\rho=-0.8$, $p<10^{-80}$) and local ($\rho=-0.8$, $p<10^{-80}$) inhibitory manipulations.
- Weakly, but significantly anticorrelated with population coupling for global manipulation of inhibition ($\rho=-0.17$, $p<10^{-5}$, Spearman rank correlation)
- Uncorrelated with population coupling for local manipulation of inhibition.

Importantly, these relationships make much of our primary results even more surprising. Namely, for right side of the peak population coupling and body coupling are anticorrelated. If spike rate is anticorrelated with both population coupling and body coupling, one would expect, naively, that population coupling would be positively correlated with body coupling. This is not what we find. Therefore, we conclude that the tendency for spike rate to be high for low body coupling cannot explain our main findings. These results are now reported in the revised manuscript in the Results section.

3. It is difficult to interpret the results from the GABA antagonist. Specifically, a systemic GABA antagonist could influence movement through direct spinal pathways and therefore any relation between cortical activity and movement response would be confounded. Moreover, while the authors conclude that their model suggests that muscimol reduces thalamic firing and therefore input to the cortex, it doesn't take into account the possibility of other bottom-up or top-down influences.

We strongly agree with the reviewer that the results of systemic administration of GABA modulators are challenging to interpret. (We note that our original motivation for systemic manipulations was related to ideas about using GABA modulators as clinically viable treatments where systemic administration is necessary. Nonetheless, the current manuscript is not addressing this point.) Therefore, we did an extensive set of new experiments (3 rats, 155 new 30 min recordings), in which we manipulated inhibition locally in motor cortex. We delivered GABA antagonists and agonists through chronically implanted microcanulas with the canula tips at the dorsal cortical surface (not penetrating) very close to the implanted electrode arrays. As expected (and unlike our systemic manipulations), local application of GABA agonist caused decreased local spike rates and local GABA antagonists tended to increase spike rates slightly (Fig 1). However, somewhat surprisingly, the changes in population coupling due to local application of muscimol were consistent with our initial findings for systemic manipulations (Fig 2). Increased local GABA agonist caused strong increases in population coupling and local GABA antagonist did not change population coupling much. Importantly, our initial primary results remained intact when considering local manipulations of inhibition.

4. It is indicated that 732 units were recorded from. However, it is not clear how many electrode arrays were these obtained from and over how many repetitions. In other words, how many 'unique' neurons were obtained?

Good question. This was difficult to address due to the electrodes that we used in our first version (200 μm inter-electrode distance). In our new experiments, we used a different style of electrode array (Buzsaki style probes from Neuronexus), which have much closer electrode spacing (20-50 μm). This allowed for much more certainty with spike sorting (see new Fig S1). And it allowed us to make some estimates of how many unique units were stable and persistent across recordings. For a subset of experiments we performed spike sorting on combined recordings (the no-drug recording and subsequent drug recording, both recorded on the same day). We found that for these cases, on average across all 64 combined recordings, 94% of units were stable and persisted from one recording to the next. To account for this fact, we added a sentence to the initial paragraph of results with a lower bound estimate on the number of unique units we studied. This number is greater than or equal to 119.

We note that we have also done substantial new analysis to show that our findings hold up to controls that do not pool units across multiple recordings (with many neurons repeated). This is discussed and reported in Fig 6.

5. Why were only 4 states evaluated in relation to movement? Also, it seems that the 0.5 second intervals used in the study are not really evaluating for spike-correlations but rather firing rate correlations. On a similar line, statistically, it is not clear how you accounted for the multiple comparisons made between all the possible body states and 4 unit states. I also find a similar issue with the STABs procedure.

Firstly, we agree with the reviewer that, with our choice of timescales for spike count time series, we are primarily studying firing rate correlations. While we could certainly have chosen finer time resolution and perhaps found additional interesting results, we thought our results were sufficiently complex and interesting as they are. Perhaps in a follow up paper, we could focus on more temporally resolved questions.

Second, regarding the 4 states... We suspect the reviewer is referring to our old approach for assessing body coupling based on mutual information. This has now been replaced with a simpler and easier to understand approach based on movement-triggered average spike rate histograms (MTASR).

Finally, regarding statistically sound multiple comparisons... We have found that, for our STABs-based approach, about 40% of neurons had "significant" body coupling BC_S . Here we defined significance as having higher BC_S than 95% of a set of surrogate control values of BC_S for each unit (1000 time-shifted controls comprised the set of

surrogate control values.) Thus, the chance-level expectation would be that 5% of neurons would be falsely deemed “significant”. This means that 40% is remarkable and highly non-random, accounting for multiple comparisons. Similarly for the other type of body coupling BC_M , we found that approximately 30% of neurons were “significant”, which is also much higher than the chance level of 5%.

6. The interpretation and conclusions of the study are not clearly stated. Even after reading the paper, it is not full apparent to me what a ‘neuron-to-body information transmission’ is. All you are seeing is really a difference in correlation between coupling and movement, but there is no time-causal analysis and no correspondence between specific body movement and neural activity to be considered as informative.

We now have attempted to state more clearly what our primary conclusions are. Our previous use of the terminology “information transmission” was chosen because our previous data analysis was based on information theoretic tools (i.e. mutual information). However, we have removed that terminology and that type of analysis from our revision. It is true that we are studying correlations among neurons and body movements, not causal relationships. Perhaps in a future study we will be able to directly cause motor cortical neurons to fire (optogenetically, for example), and monitor body movement, but for now, we report only correlative results. We hope our results are clearer now. Our primary result is most simply expressed in the title, “Strong neuron-to-body coupling implies weak neuron-to-neuron coupling in motor cortex.”

Reviewer #2 (Remarks to the Author):

This manuscript aims to determine the coupling between neurons and gross body movements in rodents spontaneously moving around a platform. Overall, the manuscript uses an interesting analytical framework and presents a potentially interesting finding. However, I am concerned that there are several major flaws in their approach. Primarily, it is really unclear if they are actually recording from a task-relevant region of motor cortex.

We appreciate the reviewer’s concern. We thank the reviewer for pointing out our oversight in not including a more detailed description of the position of our electrodes. We have now indicated the following information in our Methods (and a shorter version in Results):

... the electrode arrays were oriented such that the plane of electrodes was perpendicular to the dorsal surface and parallel to the midline. The electrodes spanned 1.4 mm in the rostrocaudal direction, centered at a point 0.5 mm caudal from bregma and 2 mm lateral from midline. The probe position was chosen deliberately to sample from neurons that are associated with a wide range of different body motions. Considering previous intracortical microstimulation studies the region we sampled is involved in many aspects of body

movement including hip flexion, trunk movements, pronation, wrist extension, elbow flexion, neck movement, and vibrissa movement (Kolb and Tees, 1990).

Although there was no trained task in our experimental design, the natural voluntary movements of the rats (grooming, rearing, more subtle posture changes, walking occasionally) are known to involve neurons in the locations we recorded from.

For example, Figure 1b shows a surprisingly quiet pattern of activation during movements?? Similarly, there are pauses in activity during increased body speed?

We could have picked an example spike raster that showed increases in spike rate with body movement. However, as we explain in our results, we found that different neurons exhibited diverse changes in firing rates related to movement; all combinations of increased, decreased, and unchanged firing were observed for both movement onset and movement cessation (Fig 4b). This is consistent with previous studies (for a few examples, see these references: (Isomura et al., 2009; Griffin et al., 2015; Peters et al., 2017). We chose an example spike raster for Fig 1b that is representative of the diverse firing patterns that we actually observed.

Obviously, the manuscript's conclusion would be very different if they were recording from a non-task modulated region.

As mentioned above (and now in our Methods and Results section), the electrodes were implanted in primary motor cortex spanning regions involved with diverse natural voluntary movements. Consistent with this, we found that many neurons showed significant modulation of firing rate around the times of spontaneous movements. This was quantified directly and called 'body coupling' in the manuscript (BC_S and BC_M). More specifically, we have found that, about 40% of neurons had "significant" BC_S . Here we defined significance as having higher BC_S than 95% of a set of surrogate control values of BC_S for each unit (1000 time-shifted controls comprised the set of surrogate control values.) Similarly for the other type of body coupling BC_M , we found that approximately 30% of neurons were "significantly" modulated in relation to body movement. Based on these numbers, it would be incorrect to say we are recording from a region that is not modulated by the body movements we study. These numbers are reported in the Results section.

The use of systemic methods for modulating inhibition also greatly diminishes enthusiasm; it makes it very difficult to interpret the results.

We strongly agree with the reviewer that the results of systemic administration of GABA modulators are challenging to interpret. (We note that our original motivation for systemic manipulations were related to ideas about using GABA modulators as clinically viable treatments where systemic administration is necessary. Nonetheless, the current

manuscript is not addressing this point.) Therefore, we did an extensive set of new experiments (3 rats, 155 new 30 min recordings), in which we manipulated inhibition locally in motor cortex. We delivered GABA antagonists and agonists through chronically implanted microcanulas with the canula tips at the dorsal cortical surface (not penetrating) very close to the implanted electrode arrays. As expected (and unlike our systemic manipulations), local application of GABA agonist caused decreased local spike rates and local GABA antagonists tended to increase spike rates slightly (Fig 1). However, somewhat surprisingly, the changes in population coupling due to local application of muscimol were consistent with our initial findings for systemic manipulations (Fig 2). Increased local GABA agonist caused increases in population coupling and local GABA antagonist did not change population coupling much. Importantly, our initial primary results remained intact when considering local manipulations of inhibition.

MAJOR

1) Very little is stated in the methods about the details of the “motor cortex” recording (i.e. coordinates, general body region targeted). The task seems to be a measure of gross movement (i.e. body speed). I imagine that the rat has to walk/run to achieve this. If the recordings were in traditional forelimb or hindlimb M1, I would expect gait cycle modulation. Demonstration of gait cycle modulation or other task parameters seems very important. At the level of resolution shown, I would also imagine dramatic increases in firing rate during bursts of movement.

As mentioned above (and now in our Methods and Results section), the electrodes were implanted in primary motor cortex spanning regions involved with natural voluntary movements, including hip flexion, trunk movements, pronation, wrist extension, elbow flexion, neck movement, and vibrissa movement (Kolb and Tees, 1990).

However, in our experiment, the rats rarely chose to walk for more than a few seconds at a time and they never chose to run. From time to time they would take a few steps to move to a new location, but the animals did not choose to walk about the observation stage for long periods. Primarily, the voluntary movements we studied in our work were more subtle than walking or running (e.g. grooming, rearing, head direction changes, more subtle posture changes). This fact precludes any demonstration of gait modulation of firing. More importantly for our study, we did demonstrate quite clearly that firing rate was modulated by the movements the animals did choose to execute. Fig 4b shows such firing rate modulation for a few example neurons. Figs 5a and b also show clearly some example neurons that were strongly associated with body movement.

We emphasize that understanding the neural basis of the movements our rats chose to execute is just as important as understanding gait and ambulation. This is particularly

true in the context of disorders that are diagnosed based on an abnormal repertoire of natural voluntary movements (e.g. autism).

We also emphasize one of the other goals of our work: to better understand the neurons in motor cortex that are not engaged with motor output. There are many neurons that are quite active, but do not fire together with body movements. This is most obvious when periods of rest are considered; when the body is entirely stationary, many neurons in motor cortex continue to fire. It is important to understand these movement-unrelated neurons and how they differ from the neurons that are strongly engaged by body movements. This goal is atypical for studies of motor cortex, but in our opinion, it is one of the strengths of our study, not a weakness.

2) Did the authors exclude periods of sleep or extended immobility? This strain is also well known to demonstrate absence seizures. Did the manipulations change that aspect?

We agree that sleep or seizures should be avoided in our study. Both these brain states often exhibit strong population-level synchrony, which would result in higher population coupling. However, based on observed LFP fluctuations during our recordings as well as our measurements of body position and posture, we concluded that the rats in our study did not sleep. Each recording session was only 30 minutes, which was not likely long enough for the animals to sufficiently relax and sleep. We were also careful to keep our doses of PTZ and Bicuculine low enough to avoid convulsive seizures. Although we did not do any special analysis to search for absence seizures, we do not think this is an issue for the following reason. One would expect that the synchrony typical of seizures to manifest as increased population coupling (at least when considering spike count time series with rather coarse time resolution, as we do in our work). One would also expect GABA antagonists (PTZ and Bicuculine) to increase the tendency of seizures. However, we did not observe an increase in population coupling with the application of GABA antagonists. In fact, we found that GABA agonist (Muscimol) strongly increased population coupling. Muscimol is likely to prevent seizures. Based on these facts, we do not think that absence seizures are playing any significant role in our observations of increased population coupling.

3) It is surprising to me that rest and movement show no difference in firing rate or coupling? What is authors explanation for this?

The reviewer is correct that we showed that resting population coupling is strongly correlated with movement population coupling. However, there is a slight tendency for higher population coupling during periods of rest. We now highlight this fact in Fig 2b with a best fit line that has slope slightly less than 1. Larger population coupling during rest is not surprising considering the typical increase in population level synchrony during rest and low-vigilance states (Harris and Thiele, 2011). The fact that the

population coupling of a neuron is almost the same for rest vs movement suggests that this property is mostly determined by the neurons place in the structure of the network in motor cortex, rather than the behavior of the animal. This argument was initially put forth by (Okun et al., 2015). They showed in sensory cortex that population coupling was correlated with how many presynaptic inputs a neuron has. Although, we can not test this idea directly in our work, our results are consistent with it.

In our original manuscript, we did not report firing rates for rest versus movement. We have now compared these quantities for each unit and found that, like population coupling, firing rates at rest are very strongly correlated with those during motion (Spearman's $\rho=0.93$, $p=0$). Moreover, we found that firing rates were typically slightly higher while the animal was at rest compared to periods of motion. We do not have an explanation for this finding, but that was not the goal of our work.

4) No spike waveforms or ISIs are shown. It is difficult to imagine that all recorded units were true single units? What was the specific criteria as I believe monotrodes were used. Multiunit recordings would of course alter the interpretation.

We agree with the reviewer that spike sorting is always imperfect. To give readers a better feeling for the quality of our spike sorting, we now include a new figure (Fig S1) showing some example spike waveforms from our recordings. We also agree that monotrode recordings are more prone to sorting errors than polytrode recordings. To ameliorate this situation, in our new experiments, we now use Buzsaki-style polytrodes (Neuronexus) together with Klusta spike sorting software. These probes and sorting software are much better for spike sorting and Klusta requires less subjective human decision (Rossant et al., 2016). We excluded units with ISIs less than 2 ms. For monotrode recordings, we set our spike cutting threshold to be deliberately liberal (“in the noise”) so that we could quantitatively compare each putative unit against noise. In this case, if the unit was not well-separated from noise based on 3 principle components and other waveform features, then it was not included. Most importantly, we found that our no-drug results were quantitatively consistent between our initial monotrode recordings and the new experiments with polytrode recordings. This strengthens our confidence that our original monotrode units were comparable in quality to our polytrode units.

5) How much acclimatization was performed? In a well-learned environment, was any motivation given to explore/walk?

We thank the reviewer for pointing out that we forgot to mention this point. We did 3 acclimatization sessions for each rat to mitigate acute stress. The rat was allowed to move about the recording stage at will for 15-20 min for each acclimatization session. We now note this in our methods section. However, as described above, in our

experiment, the rats rarely chose to walk for any extended period. From time to time they would take a few steps to move to a new location, but the animals did not choose to walk about the observation stage for long periods. The voluntary movements we studied in our work included grooming, rearing, head direction changes, and other more subtle posture changes. We emphasize that understanding the neural basis of such natural movements is just as important as understanding gait and ambulation. This is particularly true in the context of disorders that are diagnosed based on an abnormal repertoire of natural voluntary movements (e.g. autism). (More on this point below in Minor point 3.)

6) Use of a 100 surrogates seems low for the shuffling procedure.

We have increased to 1000 surrogates now. This resulted in very slight increases in the numbers of neurons with significant body coupling, but no changes in our conclusions.

MINOR

1) Provide justification for simply averaging all the markers?

Our goal was to study differences between neurons which were associated with body movement versus those that fired more independently of body movement. We wanted this distinction to cover any and all body movement, not just one particular type of movement. Therefore, we used multiple body markers to capture many aspects of movement, including the subtle ones described above (head movement, posture changes, etc.) Averaging all the marker speeds provides a convenient 1-dimensional variable that captures all movements. This simplifies our analysis (e.g. we don't have to worry about multiple comparisons). Moreover, in practice, we found that the motions of different markers were very correlated (i.e. it's hard to move one part of the body without moving other parts too). So the average marker did not behave extremely differently from any single marker.

2) Use of the 0.25 ms bin is not sufficiently explained/justified.

First, we note that we have now removed the body coupling data analysis based on mutual information, for which temporal bins were most important. Moreover, one of our current measures of body coupling (BC_S) does not require any time binning. The other measure of body coupling (BC_M) is only weakly dependent on time binning.

Early in our exploratory data analysis, we tried several different time bins including 20, 50, 100, 250, and 500 ms. Our results were not very sensitive to these changes in time bins. In the interest of not cluttering the paper with repetitive figures, we chose not to report our results for different time resolutions. We now mention this in the Methods section.

3) Please indicate why use of an untrained task is important.

The study of natural, untrained body movements is particularly important in the context of disorders that are diagnosed based on an abnormal repertoire of natural voluntary movements (e.g. autism). Indeed one of the three primary diagnostic criteria for autism is an abnormally repetitive repertoire of natural, voluntary body movements. Ultimately, a good understanding of the neural basis for such abnormal movement repertoires requires study of these behaviors under normal conditions first. Indeed, some autistic individuals perform fine on well-trained repetitive tasks like those used in standard motor-coding studies (e.g. lever press tasks). We have now added a brief justification along these lines to the manuscript (Early in Results section).

Reviewer #3 (Remarks to the Author):

The authors investigated relationship between population coupling and movement coupling in the motor cortex neurons. Their main claim is that neurons with strong population coupling are weakly coupled to body movement and conversely the ones with strong body coupling are weakly coupled to population. The claim is further supported by manipulating neuron-coupling using drugs. The question about how neuron-to-neuron coupling is related to encoding of external variables is an interesting topic and drawing a lot of attention in studies of sensory coding. Here, they aim at extending this knowledge to the motor cortex. The study is novel and data are clearly presented. Yet there are major concerns regarding the main claim of the study. Unless these concerns are addressed, the main claim is not strongly supported by the data.

1. When the authors use different ways to calculate body-coupling, the results are different in a critical way. In one condition, neuron-coupling and body-coupling are anti-correlated. In the other condition, two variables are still related significantly, but in a different way. It looks more like inverted U (or V) shape. The authors claimed that neurons with strong body coupling are weakly coupled to population, but when looking at Figure 6e, it seems more accurate to say they showed intermediate level of population coupling. Furthermore, the effect of enhanced inhibition seems to have an opposite effect on body-coupling measured in different ways. It is critical to understand the difference of these two measures.

We agree with the reviewer that the relationship between population coupling and body coupling is an inverted U, peaked shape. We now show that this is the case more clearly by plotting population coupling on a logarithmic scale for all cases. This made the rising part of the peak (left side) more apparent. (Apparently the peak is somewhat like a log-normal function in shape.) Moreover, in this revised manuscript, we decided

to discard our analysis based on mutual information. This decision was based on the difficulty of balancing dimensionality reduction against inherent low sample size biases of mutual information. In the end, we decided it was simpler and easier to understand a new measure of body coupling based on movement-triggered-average spike rate (MTASR) histograms, rather than mutual information. This new method also shows nicely the peaked relationship between body coupling and population coupling.

2. Related to the previous point, I did not understand why the authors use “standard deviation” of spike-triggered-average body speed (STABS) as a measure for body-coupling. Why not use mean (, or absolute value of mean should be more appropriate). Please explain. Please also describe time points relative to the spike used to calculate standard deviation. At time 0?

Perhaps our previous description of this measure was misleading. The standard deviation quantifies how much the STABS waveform deviates from a flat line. A unit that fires independently of body movement would have a perfectly flat STABS waveform, which would have a standard deviation of zero. A unit with a strong relationship with body movement would have a STABS waveform with a bump, like the one shown in Fig 5a, which has non-zero standard deviation. SD of the STABS waveform quantifies variation of the waveform across time. It is not evaluated at a single time point. We have now added a bit more description of this measure in the manuscript to hopefully avoid misinterpretations among future readers.

3. Anti-correlation (or distance correlation in Fig 6) between population-coupling vs. body-coupling can come from two different sources. One is across-session variation of population-coupling and body-coupling. The same neuron might show different degrees of population-coupling and body-coupling across sessions and they may co-vary. The other is across-neuron variation of population-coupling and body-coupling. The latter is what authors are interested in. Thus the contamination of the first component has to be factored out. One way is to calculate correlation between population-coupling vs. body-coupling for each session and present a histogram of this correlation. Another way is to normalize population-coupling and body-coupling for each session (standardize for each sessions, for example) before putting them together.

We agree that our previous analysis did not disentangle cross-session variability versus cross-neuron variability within sessions. We thank the reviewer for the ideas for sorting this out. We have now done the latter suggested approach. Within each recording session with at least 5 single units (this was 138 sessions total), we normalized all values of body coupling and population coupling by the median of measured values for that session. Our primary findings are robust to this standardization; we still found a clear peak in body coupling versus population coupling. This is now presented in the

new Fig 6 (panels c and d) in our manuscript. This suggests that our finding is at least partially independent of cross-session variability.

Next we sought to more directly test our findings within single recordings, without averaging across recordings, as now reported in Results and Fig 6a,b. Considering all recording sessions with at least 5 units recorded, we found that a significant number of single sessions were consistent with our primary result of a peaked relationship between BC_M and population coupling (110 out of 137 sessions, $p < 0.04$). We also found that 116 out of 137 sessions were consistent with our peaked BC_S versus population coupling result ($p < 0.003$). Here we deemed a session to be consistent with our peaked trend if it passed either of two criteria. First, we checked if a 2nd order polynomial fit had downward curvature (as a peak should). In cases where the sampled units did not pass this first criterion, we checked the slope of the fit. If the slope was positive and most of the units were to the left of the expected peak, the session was deemed consistent. (The peak location was determined based on the original pooled data. Figs 4 and 5.) If the slope was negative and most of the units were to the right of the peak, the session was deemed consistent. Statistical significance was based on repeating these checks for 1000 surrogate control datasets in which body coupling values were randomly shuffled among units.

We have also added two supplementary figures (Fig S3 and S4) showing the primary result (body coupling vs population coupling) for individual animals. We found that all 6 animals showed some tendency for a peaked function relating body coupling and population coupling, although this trend was most prominent in 4 out of the 6 rats.

4. Authors only use neurons that pass a certain criteria for body-coupling. Why not use all the neurons? Neurons with weak body-coupling are also part of data that can be used to assess relationship between body-coupling and neuron-coupling. What is the justification for using only a subset of neurons?

We agree with the reviewer that we should not exclude neurons with insignificant body coupling. We have now stopped doing this and our results still hold. However, for the reviewer's information, our original rationale for excluding neurons with insignificant body coupling was that a few neurons with very low spike rates occasionally had insignificant, but apparently strong body coupling. We wanted to avoid those cases, because they are likely to be bad estimates of body coupling and could skew our results. Now we exclude these bad cases based on their very low firing rates (fewer than 100 spikes during the whole recording). This way, we do not exclude neurons with weak body coupling, but we do exclude those that have poorly estimated body coupling.

5. Even if it is somehow justified to select a subset of neurons, choice of criteria is arbitrary. It is nice to see in Figure S2 that the main claim is insensitive to the choice of criteria. But they are not showing within condition statistics for Figure S2, which is more important. Again for figure 6, insensitivity to the choice of criteria should be presented.

We now avoid this problem by not excluding neurons with insignificant body coupling.

Minor concerns

Please cite literature claiming that Pentylenetetrazol is a GABAA antagonist.

We now cite references for the effects of PTZ, muscimol, and bicuculline.

Please describe a bit more about distance correlation coefficient.

We have now decided to use a more direct (hopefully, more transparent) analysis method to quantify the peak-like function relating body coupling and population coupling. We first find the peak of a moving average (the solid line in our plots). Then we compute a Spearman correlation coefficient for the data on the left side of the peak (should be positive for a peak relationship), and another Spearman correlation coefficient for the data on the right side of the peak (should be negative). We now report these correlations and their significance in the manuscript.

The motor cortex is a large area. Please describe the coordinates of recording.

Thanks for pointing out this oversight. We have now indicated the following information in our Methods (and a shorter version in Results):

For both groups, the electrode arrays were oriented such that the plane of electrodes was perpendicular to the dorsal surface and parallel to the midline. The electrodes spanned 1.4 mm in the rostrocaudal direction, centered at a point 0.5 mm caudal from bregma and 2 mm lateral from midline. The probe position was chosen deliberately to sample from neurons that are associated with a wide range of different body motions. Considering previous intracortical microstimulation studies the region we sampled is involved in many aspects of body movement including hip flexion, trunk movements, pronation, wrist extension, elbow flexion, neck movement, and vibrissa movement (Kolb and Tees, 1990).

Please describe the number of sessions for each drug condition.

Thanks for pointing out this oversight too. We have now indicated the following information in our Results:

In one cohort of animals (group 1, n=3 rats), we studied systemic changes in inhibition (pentylenetetrazol, PTZ at 1 mg/kg IP; or muscimol at 2 mg/kg IP). In group 1, we performed 36 no drug recordings, 18 muscimol recordings, and 19 PTZ recordings. In

another cohort (group 2, n=3 rats) we studied local changes in inhibition employing local drug infusion in motor cortex (20-1280 μ M of bicuculline or muscimol). In group 2, we performed 82 no drug recordings, 35 muscimol recordings, and 32 bicuculline recordings.

Reviewer #4 (Remarks to the Author):

The authors of this paper investigate two properties of neurons, and their interaction. First, they categorize the degree to which neurons are coupled with the population, by finding the correlation between individual units' firing patterns and the firing patterns created by summing the population of the remaining units. Second, they categorize the degree to which units are coupled to the body, by finding the correspondence between individual units' firing rates and the rats' velocity (specifically, whether movements are ending or initiating). They measured these two metrics (population coding and body coding) under three different possible intraperitoneal injections: saline (sham) injection, muscimol (GABA agonist) injection, or pentylentetrazol (GABA antagonist) injection. They report that increasing global inhibition increases population coupling and decreases body coupling. Decreasing global inhibition has the opposite effect, decreasing population coupling and increasing body coupling. They also report that during the control (saline injection) condition, individual units which are more closely linked to the population are less closely linked to the body movements, and vice-versa. They include modeling results which replicate the changes in population coupling under the different pharmacological manipulation states, under the assumption that changes in global inhibition affect both the local inhibitory units of the circuit and the overall amount of input drive to the area.

I have a number of concerns about this manuscript, which I have enumerated below. I believe the result described in this paper is potentially of interest, and could serve to inform other researchers who are attempting to link circuit properties of individual units to their role in behavior. However, I cannot recommend the paper for publication before these concerns are addressed. As written, I am insufficiently confident in the consistency and accuracy of the reported results. It is my hope that my questions and suggestions will help the authors to demonstrate their findings more rigorously and convincingly, and to better explain their significance.

1. The authors mention proposed theories as to why having units which fire in a more "coordinated" fashion might be advantageous in some circumstances (such as needing to transmit information with low noise), whereas having neurons which are more "independent" might be advantageous in others (computational efficiency). For example, in the abstract, they write: "Commands to the body might be more robustly conveyed by a strongly coupled

population, while a motor code with greater information capacity could be implemented by neurons that fire more independently” (lines 9-12). However, their results seem to go against the direction of these theories. They report that units which are more related to the body are less related to other units. This is potentially interesting, but I would like to hear the authors discuss possible reasons behind these oppositions. Without any context for these results, their significance remains unclear.

We agree with the reviewer that more discussion of our findings in relation to theoretical benefits/drawbacks of coordination improves our paper. We have now added the following paragraph to the discussion section.

What are the implications of our findings in the context of the theoretical benefits and drawbacks of coordinated firing and population coding, as discussed above in our introduction section? The peaked nature of the relationship between body coupling and population coupling suggests that motor cortex may make a compromise which balances benefits and drawbacks. The neurons that are most associated with body movement – i.e. those with peak body coupling – have small, but non-zero population coupling. They are weakly coordinated, but not completely asynchronous. This suggests that the output population of the motor cortex might balance competing needs; trading high capacity motor code (a benefit of weak coordination) for some robustness (a benefit of strong coordination). Similar trade-offs with optimal performance at intermediate levels of coordination have been observed in other aspects of cortical function as well (Alkire et al., 2008; Shew et al., 2011; Ollerenshaw et al., 2014; Gautam et al., 2015; Clawson et al., 2017). However, there remains a substantial population of neurons in motor cortex that have much higher levels of coordination among themselves, but are weakly associated with body movements. The functional role of these neurons is less clear, but could be involved with diverse “internal” operations of the cortex. Perhaps their higher level of coordination facilitates robust signal exchange with other cortical areas, but this remains to be tested.

2. This study involves three animals with chronically implanted array recording. This could mean that the number of units recorded is substantially smaller than reported: one cannot assume that each unit recorded on a given day was not already recorded on a previous day. At the same time, it is also impossible to assume that the units are stable throughout day-to-day recordings, which presents a significant quandary, as discussed by the authors in lines 206-207. Given the reported recording quantities as shown in figure S1, this indicates that the minimum “n” of units recorded is closer to 60 (maximum of 20 units recorded per day, for 3 animals). Now, it is likely that the real number is higher than 60, but also almost certainly the case that it is less than the reported 732. I have a couple of suggestions for ways to address these concerns that go beyond simply noting them in the discussion section:

We agree that we were not accurately reporting the number of unique units. We now have reported an accurate lower bound on the number of unique units (n=119) in the

first paragraph of Results. This is based on counting the largest number of units in a single recording for each rat, and then summing over all 6 rats. We also did a new spike sorting analysis on a subset (64) of combined experiments. We appended two recordings from the same day (one sham, one drug) and then did the spike sorting on the combined file. We found that units were quite stable, at least during one day. Fewer than 6% of all units were not stable across recordings.

We also agree with the reviewer that our previous manuscript did not adequately sort out the sources of variability responsible for the trends we observe. We have now considered session-to-session variability, within-session neuron-to-neuron variability, and animal-to-animal variability. We have added substantial new analysis (and new experimental data from 3 additional rats) to help sort out these questions. More details to follow below...

- Analyze datasets separately, rather than pooling all units together. Given how few animals are included in the study, I would at the very least want to see the primary analyses (e.g. figure 5) broken down by animal, to ensure that the results are consistent across animals.

We thank the reviewer for this suggestion. We have now added two supplementary figures (Fig S2 and S3) showing the primary result (body coupling vs population coupling) for individual animals. We found that all 6 animals showed some tendency for a peaked function relating body coupling and population coupling, although this trend was most prominent in 4 out of the 6 rats.

- Ideally, I would like a report of how many (if any) individual recording days feature the reported results (anti-correlation of "body coupling" and "population coupling"). This will help to ensure that a few repeatedly recorded units are not skewing the results by being included multiple times in the analysis, by requiring that the analysis be consistent within a single days' recording. Within a recording, units are presumably not repeated, and therefore will contribute only once to each recording's analysis.

To address significance of our results at the single-recording level, we have done two new analyses. First, we normalized our data to account for session-to-session variability. For this we normalized the two primary variables (body coupling and population coupling) by the median (across units) within each session. This resulted in some quantitative shifts, of course, but our primary qualitative results (peak body coupling for intermediate population coupling) were robust to this normalization, suggesting that our findings are mostly due to within-session neuron-to-neuron variability (Fig 6).

Next we sought to more directly test our findings within single recordings, without averaging across recordings. Considering all recording sessions with at least 5 units

recorded, we found that a significant number of single sessions were consistent with our primary result of a peaked relationship between BC_M and population coupling (110 out of 137 sessions, $p < 0.04$). We also found that 116 out of 137 sessions were consistent with our peaked BC_S versus population coupling result ($p < 0.003$). Here we deemed a session to be consistent with our trend if it passed either of two criteria. First, we checked if a 2nd order polynomial fit had downward curvature (as a peak should). In cases where the sampled units did not pass this first criterion, we checked the slope of the fit. If the slope was positive and most of the units were to the left of the peak, the session was deemed consistent. (The peak location was determined based on the original pooled data. Figs 4 and 5.) If the slope was negative and most of the units were to the right of the peak, the session was deemed consistent. Statistical significance was based on repeating these checks for 1000 surrogate control datasets in which body coupling values were randomly shuffled among units. These results are now reported in Fig 6 and in the text following Fig 5.

3. In the legend for figure 5 (lines 440-442), the authors report “Among those neurons with significant body coupling (colored), high population coupling was found only for neurons with low body coupling.” This has led me to be concerned that the authors’ method of assessing the anti-correlation between body coupling and population coupling only took into account units with significant body coupling. Is this the case? If so, they should change their analysis to take into account all units: presumably those with non-significant coupling should be those with, essentially, the “lowest” coupling values, and thus these units, too, are important for the analysis. Are units with no significant body coupling more likely to have large population coupling and vice-versa?

We agree with the reviewer that we should not have excluded neurons with insignificant body coupling. We have now stopped doing this and found that our results still hold. However, for the reviewer’s information, our original rationale for excluding neurons with insignificant body coupling was that a few neurons with very low spike rates occasionally had insignificant, but apparently strong body coupling. We wanted to avoid those cases, because they are likely to be bad estimates of body coupling and could skew our results. Now we exclude these bad cases based on their very low firing rates (fewer than 100 spikes during the whole recording). This way, we do not exclude neurons with weak body coupling, but we do exclude those that have poorly estimated body coupling.

4. How stable are the metrics of population coupling and body coupling? For population coupling, the authors report that there is not a significant change between the coupling during movement or at rest. However, there seems to be little to no correspondence between the population coupling values (and the body coupling values) recorded from units on the same electrode and different days. This could be explained either by different units being recorded on

the same electrode, or by drift in these values over time. This has led me to be concerned about how stable these metrics are within a recording session. If you cut the session into a first half and a second half, are the population coupling values and the body coupling values similar? This could also be something which could be assessed via the model (for population coupling): do the model population coupling values remain stable over time?

We agree with the reviewer that a comparison between the first and second halves of each recording is a nice way to assess stability of population coupling and body coupling. We have now done this and shown population coupling is quite stable on the time scale of the recordings. Population coupling for the first half of each recording was strongly correlated with that during the second half (Pearson $\rho=0.80$, $p=0$). Similarly, BCM for the first half of each recording was strongly correlated with that during the second half (Pearson $\rho=0.74$, $p=0$). And BCS for the first half of each recording was strongly correlated with that during the second half (Pearson $\rho=0.62$, $p=0$).

Furthermore, the model is a good way to assess how likely the population coupling as measured by a subset of the population matches the population coupling using the whole population. If you measure population coupling with the entire 1000-unit population, compared with the 20-unit subpopulations, how correlated are these measurements?

We agree that the model could give a possible answer to this question, but in the end, we felt that our single session assessments of the peaked trend discussed above is better.

5. In how much agreement are the STABS metric and the mutual information metric of body coupling? Are units which tend to be body coupled using one metric also body coupled using the other metric?

We have now shown in Fig 5c that our two different metrics for measuring body coupling are correlated but not identical. We note that in our revision, we have discarded the mutual information based approach in favor of the movement-triggered-average spike rate (MTASR) approach, which is less sensitive to finite sample size biases.

6. The authors use different thresholds to assess significance for different analyses. For example, the mutual information between the unit firing rate and the “starting/stopping movement” times had to exceed 90% of the shuffled values in order to be considered significant, whereas the STABS metric only needed to exceed 60% of the shuffled values. Furthermore, population coupling is reported in degree but not in significance, even though it seems that a “shuffled spike time” analysis similar to that for STABS would also function to determine the significance

of the population coupling. I request that they always note why a particular threshold was chosen.

We now present all values of body coupling. We no longer exclude those below a threshold level of significance. One exception is that we now exclude units that had very low spike rates (fewer than 100 spikes during a recording), because these units had unreliable body coupling values. We do not have a principled reason for choosing 100 spikes per recording, however we did try other cutoffs (50, 200, 500 spikes per recording) and found that our results did not change much, which we now point out in our Online Methods section.

7. The pharmacological conditions were analyzed together when comparing population coupling to body coupling using mutual information, but analyzed separately when using STABS. Either the two metrics (STABS and mutual information) should be analyzed as similarly as possible, or a good justification should be provided. In addition, the primary result (anti-correlation of body coupling and population coupling) changes depending on the metric of body coupling which is used. Either there is a simple anti-correlation between population coupling and body coupling, or there is some “peak” (which also shifts depending on the pharmacological condition, and presumably disappears in the “reduced inhibition” conditions, given that the authors did not show this condition’s results). This lack of correspondence is concerning, and I would like the authors to give more thought as to why it might have occurred, and what it means for their results.

We agree... We have now shown that both our approaches for measuring body coupling (BC_M and BC_S) are consistent with a peaked relationship between body coupling and population coupling. We have also now separated the drug and no-drug conditions in all the body coupling vs population coupling plots.

Minor questions/comments:

- Sham and injection experiments were always done on the same days (and presumably in the same order). Is there an effect of the fact that the experimental conditions were always (presumably) the second experimental session of the day? Are the same units recorded on sham and non-sham experiments on the same day?

To answer this question, for a subset (64) of our experiments, we did our spike sorting on two combined recordings (sham + drug). We found that the vast majority of units were stable and present in both the sham and drug recordings. We have used this information to provide a lower estimate of the total number of unique units we have measured in our data set. This is now reported in the 1st paragraph of Results.

Because of the time we expect for the drug (non-sham) conditions to return to a baseline, it was not feasible to do drug condition followed by a sham condition on the same day.

- In figure 4c, are these six different units, or three units whose “increased speed” and “decreased speed” tuning are both shown?

Fig 4b (previously 4c) shows 6 different units. However, relevant to the reviewer’s question, we now show that the body coupling based on movement onset events is strongly correlated with that based on movement cessation events. Thus, the units that are strongly modulated by movement initiation are also strongly modulated by movement cessation.

- Why is figure 5b shown in logarithmic scale, while figure 5a is shown in linear scale?

We have now shown all population coupling results on log scale axes. This was important for revealing the peaked shape of the body coupling vs population coupling relationship. This suggests that the relationship is shaped something like a lognormal function.

- In line 128, “can accounts” should be “can account.”

Fixed. Thanks.

- Because the STABS metric is assessed via standard deviation, it is preferentially searching for units with a strong correlation with either increases in body speed or decreases in body speed. For example, the STABS metric would not flag a unit as “body coupled” if the unit always fired when the rat was running at a high speed - The STABS metric would show a flat line that happened to be a higher speed than the shuffled trials, which would be normalized away. The authors comment that this metric “is not biased toward the somewhat arbitrarily defined events of movement initiation and termination considered above” (159-161). They should make clear the fact that STABS is also primarily searching for units which tend to fire around speeding up or slowing down events.

We agree with the reviewer, that our STABS metric would miss a neuron that only began to fire once an animal reached a high constant speed. However, in our experience, the vast majority of the natural voluntary movements of the animal were brief in duration (~1 s or less in duration). Typical movements included brief changes in position, changes in posture, rearing, grooming, etc. Our animals did not choose to move about the stage continuously. Continuous high velocity scenarios, like the reviewer describes, occurred rarely if at all. Thus, we think that our STABS metric and our MTASR metric are likely to account for the vast majority of the movements that our

animals made. We have now noted this in the paragraph of the Results where we define the STABS metric.

References cited in this reply to reviewers

Alkire MT, Hudetz AG, Tononi G (2008) Consciousness and anesthesia. *Science* 322:876–880.

Clawson WP, Wright NC, Wessel R, Shew WL (2017) Adaptation towards scale-free dynamics improves cortical stimulus discrimination at the cost of reduced detection Hilgetag CC, ed. *PLOS Comput Biol* 13:e1005574.

Gautam SH, Hoang TT, McClanahan K, Grady SK, Shew WL (2015) Maximizing Sensory Dynamic Range by Tuning the Cortical State to Criticality. *PLOS Comput Biol* 11:e1004576.

Griffin DM, Hoffman DS, Strick PL (2015) Corticomotoneuronal cells are “functionally tuned.” *Science* (80-) 350:667–670.

Harris KD, Thiele A (2011) Cortical state and attention. *Nat Rev Neurosci* 12:509–523.

Isomura Y, Harukuni R, Takekawa T, Aizawa H, Fukai T (2009) Microcircuitry coordination of cortical motor information in self-initiation of voluntary movements. *Nat Neurosci* 12:1586–1593.

Kolb B, Tees RC (1990) *The Cerebral Cortex of the Rat*. Cambridge, MA: MIT Press.

Okun M, Steinmetz N a., Cossell L, Iacaruso MF, Ko H, Barthó P, Moore T, Hofer SB, Mrcic-Flogel TD, Carandini M, Harris KD (2015) Diverse coupling of neurons to populations in sensory cortex. *Nature*.

Ollerenshaw DR, Zheng HJ V, Millard DC, Wang Q, Stanley GB (2014) The adaptive trade-off between detection and discrimination in cortical representations and behavior. *Neuron* 81:1152–1164.

Peters AJ, Lee J, Hedrick NG, O’Neil K, Komiyama T (2017) Reorganization of corticospinal output during motor learning. *Nat Neurosci* 20:1133–1141.

Rossant C, Kadir SN, Goodman DFM, Schulman J, Hunter MLD, Saleem AB, Grosmark A, Belluscio M, Denfield GH, Ecker AS, Tolias AS, Solomon S, Buzsáki G, Carandini M, Harris KD (2016) Spike sorting for large, dense electrode arrays. *Nat Neurosci* 19:634–641.

Shew WL, Yang H, Yu S, Roy R, Plenz D (2011) Information Capacity and Transmission Are Maximized in Balanced Cortical Networks with Neuronal Avalanches. *J Neurosci* 31:55–63.

Reviewers' Comments:

Reviewer #1:

Remarks to the Author:

The authors have done a truly commendable job at addressing all of my major concerns. In particular, they had performed new experiments in three additional animals using a more localized technique. They have also clarified several points and amended the manuscript accordingly. I have no additional concerns at this point, and recommend publication.

Reviewer #3:

Remarks to the Author:

The main claim of this study is somewhat changed from the previous version and now it is that neurons' population coupling and movement coupling is related and the relationship is in an inverted U/V shape: neurons involved in external body movement have an intermediate level of population coupling. Overall this relationship seems to be weak and noisy. I also have some concerns on statistical analyses. I listed major concerns below.

- 1) This relationship seems to be weak and noisy. One of the main figures to support this claim is Figure 5. But in the no drug condition and in the enhanced inhibition condition, one side of the slope is not significant. Thus in these cases, authors cannot claim that the relationship has a significant peak.
- 2) Any data have a peak (maximum), and the left side of the peak tends to have a positive slope and the right side of the peak tends to have a negative slope. The authors' claim might be affected by this trivial bias. Please describe clearly it is not the case. Especially a clear and detailed description of the analysis procedure is critical. Is this bias taken into account in a shuffle test? It is not clear at which point shuffling is performed. Shuffling of population coupling should be performed first, then the following procedure should be performed exactly the same way as performed on the real data, such as choosing bins, defining a peak and performing correlation analyses for the left side and the right side.
- 3) For the analysis of Figure 6a,b, as a first criterion, the authors used a negative coefficient for the quadratic term in the second order polynomial fit. But if a peak of the fit is not located within the range of data, then it is not consistent with the inverted U/V relationship. The number of "consistent" single recordings may be significantly different from the chance level, but I am not sure what this is proving if some recordings are defined as "consistent" even though they are actually not consistent. Furthermore in the second criterion, what is the definition of "most" of the units were left/right of the peak? More than half?, More than 90%? Lastly, the peak is defined using a pooled data across sessions. In a shuffling test, the peak should be re-defined using a pooled shuffling data, not from the real data. If not, please re-analyze. If it is the case, please state so.
- 4) In many analyses, the location of the peak is probably defined as a peak in the binned data. So the result is probably sensitive to choice of bins. How are the bin sizes chosen?
- 5) In a local drug injection experiment, multiple concentrations were tested. But in the result, they just divided into "weak" and "strong". What are the definitions of "weak" and "strong". Furthermore, in the analysis of relationship between population coupling and body coupling, the authors seem to use only the "strong" conditions. What is the justification for this?

Reviewer #4:

Remarks to the Author:

The authors have added substantial new analyses and experiments to their paper which does improve it considerably. In particular, they have added local inhibition and excitation experiments to complement their prior global pharmacological manipulations. They have also included analyses which show that data collected in single sessions and single animals agrees with the global result, reducing concerns about the effects being driven either by combining data across animals or by an effect in a subset of the animals. I think that these new experiments and analyses are a step in the right direction. However, I retain several methodological and interpretation related concerns which temper my enthusiasm for this paper. I therefore feel that I cannot, at this time, recommend this paper for publication in Nature Communications.

Major concerns:

1. I remain skeptical of the authors' definition of body coupling. In particular, the authors' methods select for neurons whose activity is correlated with increases or decreases in global body speed. Their methods are well-designed for testing for this parameter; however, it is likely that they are missing large classes of "body coupling" type neurons. For example, a neuron whose tuning was related to a single body part could easily be missed in their analysis. Thus, it is a bit hard to say that the neurons that have low body coupling in their analysis are unlikely to be "output neurons", as suggested in their discussion. This dramatically tempers the impact of the paper, as it is hard to make a strong statement about whether all neurons tend to have this "peaked" relationship between body coupling and population coupling, or just those units which have a very particular relationship with accelerations and decelerations in whole-body movements. Unfortunately, I do not see a simple way to improve this analysis with the experimental procedure they are currently using. The advantage of using un-trained movements for this paper is, as the authors suggest, the potential variety of movements that can be studied (and also ease of running the experiments). However, the disadvantage is that it can be very difficult to tease out the relationship between the units and the eventual output – indeed, the authors do not make use of the complexity of movements that the animals display, instead needing to resort to considering only average body velocity. This gives them the statistical power that they need to connect neurons to body movements, but almost certainly "averages out" plenty of interesting relationships between the neurons and more complex or more specific movement parameters.

2. The authors use a time bin for assessing population coupling which is much longer than the time bin used in Okun et al, 2015, the paper which initially described the phenomenon of variable population coupling between units. In the Okun paper, population coupling is calculated on the order of milliseconds (firing rates are placed in 1 ms bins, and convolved with a 12-ms half-width gaussian). The definition of population coupling as used by Okun and colleagues is likely to pick up more strongly on "noise correlations" and is suggested in that paper to strongly reflect local connectivity. I worry that a 250-ms bin size may smooth out those short-timescale correlations and instead reflect similarity of tuning between the units and the population. The authors of this paper note that selection of different time bins had little effect on their results, but I would like to see a supplementary figure or a sub-panel showing that the population coupling values of neurons found in this paper are insensitive to selections of smaller time bins. A comparison of the average firing rates here to those from studies of visual cortex might also help to build intuition for a reasonable range of bin sizes.

3. I am skeptical of the authors' statistical analysis of the peaked relationship between body coupling and population coupling. To perform this analysis, they examine whether the points to the left of the peak have a significant "positive" slope and the points to the right of the peak have a significant "negative" slope. However, the selection of the peak point can change for each analysis, and is selected to be the point that maximizes the likelihood of finding a significant positive and negative

slope. This is problematic for two reasons:

a. By selecting a different "peak" point for each analysis, the authors bias themselves toward finding a significant peak. By definition, a peak will have lower values to the left and to the right, thus tending to cause a positive and negative correlation on either side, even if the peak itself is spurious.

b. On line 185 the authors note that "Interestingly, for local enhancement of inhibition all the neurons were on the left side of the peak..." This suggests that results of a purely positive correlation or a purely negative correlation would also "support" the hypothesis of a peaked relationship, despite the fact that in these cases, there is not a peak present in the data.

I suggest that the authors try a less-biased form of statistical analysis for the peaked nature of their data. Perhaps some form of bootstrap method could be applied, in which each cell's body coupling is paired with a random other cell's population coupling (thus keeping the two distributions the same, but interfering with any potential relationship between them). I also note that it is possible that the authors are doing this already, in which case I apologize for my confusion and request that they clarify how this analysis is performed. The authors do mention performing shuffle controls for their p-values in the methods section, but the paper would benefit from more details in the methods of how the shuffles were performed for this particular analysis. In particular, was the shuffle performed only for the points to the "left" and "right" of the peak, or was the shuffle performed for all points and then peaked-ness reassessed?

Minor issues:

Line 132: "effecting" should be "affecting"

Line 279-281: The phrase "Future experiments in which output neurons are 'unnaturally' coopted for control of brain machine interface devices would provide a more direct test of this intriguing idea" is unclear. Is the suggestion that assigning units randomly to be the "output" units of a BMI would cause them to reduce their population coupling?

Reviewer #1 (Remarks to the Author):

The authors have done a truly commendable job at addressing all of my major concerns. In particular, they had performed new experiments in three additional animals using a more localized technique. They have also clarified several points and amended the manuscript accordingly. I have no additional concerns at this point, and recommend publication.

We appreciate the support and positive feedback.

Reviewer #3 (Remarks to the Author):

The main claim of this study is somewhat changed from the previous version and now it is that neurons' population coupling and movement coupling is related and the relationship is in an inverted U/V shape: neurons involved in external body movement have an intermediate level of population coupling. Overall this relationship seems to be weak and noisy. I also have some

concerns on statistical analyses. I listed major concerns below.

We agree with the reviewer that the peaked relationship is fairly noisy. However, as we describe in more detail below, we have now developed a much better analytical approach for assessing the statistical significance of a noisy peaked relationship. Using this new method, we are now able to make our conclusions with more confidence.

1) This relationship seems to be weak and noisy. One of the main figures to support this claim is Figure 5. But in the no drug condition and in the enhanced inhibition condition, one side of the slope is not significant. Thus in these cases, authors cannot claim that the relationship has a significant peak.

With our improved method for assessing significance, we no longer depend on significance of correlations on either side of the peak. Here is a description of the new method as it is now written in the Methods section of the paper:

To assess the statistical significance of the non-monotonic peaked relationship between body coupling and population coupling, we performed the following several steps. First, we fit the data with a 4th-order polynomial $f(x)=a+bx+cx^2+dx^3+ex^4$, where f represents body coupling, x represents the logarithm of population coupling, and a , b , c , d , and e are fit constants. Separate fits were done for each drug condition and for the three types of body coupling (BC_M , BC_S , and BC_D). We used the peak of this fit function to identify the value of population coupling with peak body coupling. Next we computed a Spearman correlation coefficient ρ_{left} for the points to the left of the peak and another ρ_{right} for the points to the right of the peak. First, we note that in all cases, ρ_{left} was positive and ρ_{right} was negative, as expected for a peak. Then we repeated this process for 1000 surrogate shuffled datasets for which population coupling values were randomized relative to the body coupling values. This shuffling procedure was followed by the same three steps: finding a peak using a fit polynomial, calculating ρ_{left} , and calculating ρ_{right} . The p value reported in the main text is the probability of finding a surrogate shuffled dataset with a greater ρ_{left} and a smaller ρ_{right} compared with our actual measured ρ_{left} and ρ_{right} .

2) Any data have a peak (maximum), and the left side of the peak tends to have a positive slope and the right side of the peak tends to have a negative slope. The authors' claim might be affected by this trivial bias. Please describe clearly it is not the case. Especially a clear and detailed description of the analysis procedure is critical. Is this bias taken into account in a shuffle test? It is not clear at which point shuffling is performed. Shuffling of population coupling should be performed first, then the following procedure should be performed exactly

the same way as performed on the real data, such as choosing bins, defining a peak and performing correlation analyses for the left side and the right side.

We agree that our previously reported method needed improvement. As the reviewer suggested, our improved method now is based on applying the same procedures to shuffled data and real data, as described in our response to the last point above.

3) For the analysis of Figure 6a,b, as a first criterion, the authors used a negative coefficient for the quadratic term in the second order polynomial fit. But if a peak of the fit is not located within the range of data, then it is not consistent with the inverted U/V relationship. The number of “consistent” single recordings may be significantly different from the chance level, but I am not sure what this is proving if some recordings are defined as “consistent” even though they are actually not consistent. Furthermore in the second criterion, what is the definition of “most” of the units were left/right of the peak? More than half?, More than 90%? Lastly, the peak is defined using a pooled data across sessions. In a shuffling test, the peak should be re-defined using a pooled shuffling data, not from the real data. If not, please re-analyze. If it is the case, please state so.

We agree with all the reviewer’s suggested improvements here. After making these improvements, we still found a statistically significant number (although a smaller number than before these improvements) of single recordings that are consistent with our pooled results. We have now included a criterion for consistency that requires the fit peak be located within $\pm 10\%$ of the peak location for the pooled data. We have now clarified in the revised methods what was meant by “most of the units” being right or left of the peak. For example, if “most of the units” were right of the peak, this meant, more precisely, that the majority (more than half) of the range (min to max) of population coupling for those units was to the right of the peak. Finally, we also redefined the peak based on pooled data after shuffling the pooled data.

4) In many analyses, the location of the peak is probably defined as a peak in the binned data. So the result is probably sensitive to choice of bins. How are the bin sizes chosen?

We thank the reviewer for another good idea. In all cases where we sought a peak from pooled data, we have now located peaks based on the peak of a best-fit fourth order polynomial. Thus, finding the peak is now totally independent of any choice of bins used for visualizing the average of the scattered points. Now the bins can be thought of as simply a visual aid, not relevant for any of our actual data analytic calculations.

5) In a local drug injection experiment, multiple concentrations were tested. But in the result, they just divided into “weak” and “strong”. What are the definitions of “weak” and “strong”. Furthermore, in the analysis of relationship between population coupling and body coupling, the authors seem to use only the “strong” conditions. What is the justification for this?

We thank the reviewer for pointing out this oversight. We now explain the division of the experiments into two groups as follows (now in Pharmacology section of Methods).

We found that many of the effects of the local drug manipulations were quite small for the lower drug concentrations. To obtain a clearer view of drug effects, we divided the experiments into two groups for some of our results. The “strong” group included concentrations 320, 640, and 1280 μM ; the “weak” group included 20, 40, 80, and 160 μM .

Reviewer #4 (Remarks to the Author):

The authors have added substantial new analyses and experiments to their paper which does improve it considerably. In particular, they have added local inhibition and excitation experiments to complement their prior global pharmacological manipulations. They have also included analyses which show that data collected in single sessions and single animals agrees with the global result, reducing concerns about the effects being driven either by combining data across animals or by an effect in a subset of the animals. I think that these new experiments and analyses are a step in the right direction. However, I retain several methodological and interpretation related concerns which temper my enthusiasm for this paper. I therefore feel that I cannot, at this time, recommend this paper for publication in Nature Communications. We thank the reviewer for the positive comments and constructive suggestions for further improvements.

Major concerns:

1. I remain skeptical of the authors' definition of body coupling. In particular, the authors' methods select for neurons whose activity is correlated with increases or decreases in global body speed. Their methods are well-designed for testing for this parameter; however, it is likely that they are missing large classes of “body coupling” type neurons. For example, a neuron whose tuning was related to a single body part could easily be missed in their analysis. Thus, it is a bit hard to say that the neurons that have low body coupling in their analysis are unlikely to be “output neurons”, as suggested in their discussion. This dramatically tempers the impact of the paper, as it is hard to make a strong statement about whether all neurons tend to have this “peaked” relationship between body coupling and population coupling, or just those units which have a very particular relationship with accelerations and decelerations in whole-body movements. Unfortunately, I do not see a simple way to improve this analysis with the experimental procedure they are currently using. The advantage of using un-trained movements for this paper is, as the authors suggest, the potential variety of movements that can be studied (and also ease of running the experiments). However, the disadvantage is that it can be very difficult to tease out the relationship between the units and the eventual output –

indeed, the authors do not make use of the complexity of movements that the animals display, instead needing to resort to considering only average body velocity. This gives them the statistical power that they need to connect neurons to body movements, but almost certainly “averages out” plenty of interesting relationships between the neurons and more complex or more specific movement parameters.

We agree with the reviewer that the results in our previous version left open the possibility that we were averaging out important details in the movements. To improve upon this aspect of our work, we have now added a new type of data analysis to our revision that we feel successfully generalizes our primary finding to account for many more detailed aspects of body movement. These new results are now presented in Fig 7. We acknowledge that our 8 motion tracking beads cannot capture all aspects of body movement, but our new analysis demonstrates that our primary results hold for many types of detailed body movements, not just the average body speed.

We describe the new analysis in detail in the paper, but in brief, we are now accounting for 27 aspects of motion including: 1) center-of-mass speed, 2) right-left turning angular speed, 3) up-down rearing angular speed, 4-27) 3 components of velocity for each of the 8 tracking beads, measured relative to a “body vector” aligned with the rat’s rostral-caudal body axis. As the reviewer anticipated, we found that many units fired in relation to rather specific detailed motions. So we defined a third type of body coupling based on the “preferred” type of body movement for each unit. This way, a unit will not be assigned low body coupling if it only fires in relation to a subtle motion that is not reflected in the average body speed.

2. The authors use a time bin for assessing population coupling which is much longer than the time bin used in Okun et al, 2015, the paper which initially described the phenomenon of variable population coupling between units. In the Okun paper, population coupling is calculated on the order of milliseconds (firing rates are placed in 1 ms bins, and convolved with a 12-ms half-width gaussian). The definition of population coupling as used by Okun and colleagues is likely to pick up more strongly on “noise correlations” and is suggested in that paper to strongly reflect local connectivity. I worry that a 250-ms bin size may smooth out those short-timescale correlations and instead reflect similarity of tuning between the units and the population. The authors of this paper note that selection of different time bins had little effect on their results, but I would like to see a supplementary figure or a sub-panel showing that the population coupling values of neurons found in this paper are insensitive to selections of smaller time bins. A comparison of the average firing rates here to those from studies of visual cortex might also help to build intuition for a reasonable range of bin sizes.

We agree that this point was inadequately addressed in our last version. We have now done a more systematic study of how our primary results depend on temporal resolution. We now

directly compare population coupling measured with 250 ms resolution to that measured at three finer resolutions: 100, 50, and 10 ms. These new results are now presented in Fig S5.

We found that population coupling was highly correlated across these different temporal scales. Most importantly, we found that our primary results relating body coupling to population coupling were largely robust to these changes in temporal scale (Fig S5). We note that our analyses of body coupling did not include a choice of time bin size.

3. I am skeptical of the authors' statistical analysis of the peaked relationship between body coupling and population coupling. To perform this analysis, they examine whether the points to the left of the peak have a significant "positive" slope and the points to the right of the peak have a significant "negative" slope. However, the selection of the peak point can change for each analysis, and is selected to be the point that maximizes the likelihood of finding a significant positive and negative slope. This is problematic for two reasons:

a. By selecting a different "peak" point for each analysis, the authors bias themselves toward finding a significant peak. By definition, a peak will have lower values to the left and to the right, thus tending to cause a positive and negative correlation on either side, even if the peak itself is spurious.

b. On line 185 the authors note that "Interestingly, for local enhancement of inhibition all the neurons were on the left side of the peak..." This suggests that results of a purely positive correlation or a purely negative correlation would also "support" the hypothesis of a peaked relationship, despite the fact that in these cases, there is not a peak present in the data.

I suggest that the authors try a less-biased form of statistical analysis for the peaked nature of their data. Perhaps some form of bootstrap method could be applied, in which each cell's body coupling is paired with a random other cell's population coupling (thus keeping the two distributions the same, but interfering with any potential relationship between them). I also note that it is possible that the authors are doing this already, in which case I apologize for my confusion and request that they clarify how this analysis is performed. The authors do mention performing shuffle controls for their p-values in the methods section, but the paper would benefit from more details in the methods of how the shuffles were performed for this particular analysis. In particular, was the shuffle performed only for the points to the "left" and "right" of the peak, or was the shuffle performed for all points and then peaked-ness reassessed?

We agree that this part of our analysis needed an overhaul. We have now done this. We have devised a totally new and improved method for assessing the significance of a peaked relationship. This new method is now used throughout the paper wherever we check for a peaked relationship for pooled data. The method is described in Methods now as follows:

To assess the statistical significance of the non-monotonic peaked relationship between body coupling and population coupling, we performed the following several steps. First, we fit the data with a 4th-order polynomial $f(x)=a+bx+cx^2+dx^3+ex^4$, where f represents body coupling, x represents the logarithm of population coupling, and a , b , c , d , and e are fit constants. Separate fits were done for each drug condition and for the three types of body coupling (BC_M , BC_S , and BC_D). We used the peak of this fit function to identify the value of population coupling with peak body coupling. Next we computed a Spearman correlation coefficient ρ_{left} for the points to the left of the peak and another ρ_{right} for the points to the right of the peak. First, we note that in all cases, ρ_{left} was positive and ρ_{right} was negative, as expected for a peak. Then we repeated this process for 1000 surrogate shuffled datasets for which population coupling values were randomized relative to the body coupling values. This shuffling procedure was followed by the same three steps: finding a peak using a fit polynomial, calculating ρ_{left} , and calculating ρ_{right} . The p value reported in the main text is the probability of finding a surrogate shuffled dataset with a greater ρ_{left} and a smaller ρ_{right} compared with our actual measured ρ_{left} and ρ_{right} .

Minor issues:

Line 132: “effecting” should be “affecting”

Fixed. Thanks.

Line 279-281: The phrase “Future experiments in which output neurons are ‘unnaturally’ coopted for control of brain machine interface devices would provide a more direct test of this intriguing idea” is unclear. Is the suggestion that assigning units randomly to be the “output” units of a BMI would cause them to reduce their population coupling?

Here the idea was that choosing neurons with low population coupling to control a machine might result in better control of the machine. And choosing neurons with high population coupling might result in worse control. We have now revised this bit of speculation in the conclusion section to make this clearer.

Reviewers' Comments:

Reviewer #3:

Remarks to the Author:

My main concern regarding the weak and noisy U/V relationship between population coupling and body coupling was not eliminated by the authors' responses. Because this U/V relationship is one of the main conclusions of this paper, this has to be robust and clear. At this point, I don't think the manuscript meets the criteria for publication in Nature Communications.

(1) Why do authors use a 4th order polynomial fit? This seems arbitrary and overly complicated.

(1a) Does the fit produce only one peak within the range of data in all the conditions?? I recommend the authors provide actual 4th order polynomial fit in the main figures so that readers could judge whether it makes sense to use this high-order polynomial fit. Then indicate the peak in the figure and provide correlation coefficients for the left side and the right side in the figure legend.

(1b) Definition of the peak is not clear because it could produce multiple peaks. It is also possible that there is no peak within the range of data or there is a peak but there are other data points bigger than the peak. Please describe how the peak is defined in any possible situations. Otherwise it is hard to assess the validity of this analysis.

(1c) To produce a p-value, the authors ask how many shuffled data among 1000 shuffles have higher p_{Left} and p_{Right} (with a sign flip for p_{Right}). Because they require two conditions to be simultaneously met, the p-value generated here is a generous estimate, if I understand correctly. If this is the case, it should be corrected.

(2) An alternative way to find and prove existence of a significant peak is to search for a peak in an exhaustive way, instead of relying on a fit. In this analysis, first divide data into two by selecting a criterion value. Try all possible ways to divide data into two. (If there are 1000 neurons, there should be 999 ways to divide data into two. But probably dividing data into 1 and 999 neurons is not good for further analyses. So start from 20 and 980, for example, so that reliable estimates of both p_{Left} and p_{Right} could be obtained). Then define a "peakiness" index as the smaller of p_{Left} and $-p_{\text{Right}}$. A division that provides the maximum peakiness index is the location of the peak. Repeat this for 1000 shuffled data and obtain 1000 maximum peakiness indices. Obtain a p-value by comparing the maximum peakiness index of real data to the distribution of maximum peakiness indices of shuffled data. Authors could apply this to Figure 4d, 5d, 6c, 6d and 7b. (In the figures, they should provide both a left side fit and a right side fit.).

Reviewer #4:

Remarks to the Author:

The authors have revised their manuscript in response to my additional concerns, significantly improving their statistics and data analysis. This increases my confidence in their findings, and I am now happy to recommend this paper for publication.

One final suggestion (though by no means required) would be to consider modifying the title of the paper. Given that the paper describes a peaked relationship between population coupling and body coupling, the title (while not strictly inaccurate) could potentially lead to confusion.

Reviewer #3 (Remarks to the Author):

My main concern regarding the weak and noisy U/V relationship between population coupling and body coupling was not eliminated by the authors' responses. Because this U/V relationship is one of the main conclusions of this paper, this has to be robust and clear. At this point, I don't think the manuscript meets the criteria for publication in Nature Communications.

We thank the reviewer for carefully considering how to improve our test for statistical significance of a noisy peaked relationship. Based on the reviewer's suggestions, we have now developed a new, more strict test for significance. This new method is highlighted in the methods section and described below.

(1) Why do authors use a 4th order polynomial fit? This seems arbitrary and overly complicated.

Although we are no longer using this fitting method in our revision, we note that we previously found a 4th order polynomial fit better than lower orders, because our noisy peaked data is sometimes rather skewed, asymmetrical around the peak.

(1a) Does the fit produce only one peak within the range of data in all the conditions?? I recommend the authors provide actual 4th order polynomial fit in the main figures so that readers could judge whether it makes sense to use this high-order polynomial fit. Then indicate the peak in the figure and provide correlation coefficients for the left side and the right side in the figure legend.

There were not two peaks in our previous fits, but since we switched to a different method for peak testing, we have not added the fit functions in the figures.

(1b) Definition of the peak is not clear because it could produce multiple peaks. It is also possible that

there is no peak within the range of data or there is a peak but there are other data points bigger than the peak. Please describe how the peak is defined in any possible situations. Otherwise it is hard to assess the validity of this analysis.

Our previous method did not produce two peaks, but since we switched to a different method for peak testing, we have not included new figures to prove this.

We now specify exactly how a putative peak is located in our methods section. Notably, we require that valid choices for peak locations have positive ρ_{Left} and negative ρ_{Right} . If no valid choices of a peak location meet this criterion then our algorithm simply stops, and we conclude there is no peak. This happened frequently for shuffled data, but never for the actual data.

(1c) To produce a p-value, the authors ask how many shuffled data among 1000 shuffles have higher ρ_{Left} and ρ_{Right} (with a sign flip for ρ_{Right}). Because they require two conditions to be simultaneously met, the p-value generated here is a generous estimate, if I understand correctly. If this is the case, it should be corrected.

We agree that our previous method was somewhat liberal, resulting in lower p values than our new method (see more detail below).

(2) An alternative way to find and prove existence of a significant peak is to search for a peak in an exhaustive way, instead of relying on a fit. In this analysis, first divide data into two by selecting a criterion value. Try all possible ways to divide data into two. (If there are 1000 neurons, there should be 999 ways to divide data into two. But probably dividing data into 1 and 999 neurons is not good for further analyses. So start from 20 and 980, for example, so that reliable estimates of both ρ_{Left} and ρ_{Right} could be obtained). Then define a "peakiness" index as the smaller of ρ_{Left} and $-\rho_{\text{Right}}$. A division that provides the maximum peakiness index is the location of the peak. Repeat this for 1000 shuffled data and obtain 1000 maximum peakiness indices. Obtain a p-value by comparing the maximum peakiness index of real data to the distribution of maximum peakiness indices of shuffled data. Authors could apply this to Figure 4d, 5d, 6c, 6d and 7b. (In the figures, they should provide both a left side fit and a right side fit.).

We have now adopted a method based on (but not identical to) this nice suggestion from the reviewer. We no longer use a polynomial fit to find the peak location. Instead, we use the reviewer's idea of scanning through all possible choices of a peak location (excluding 30 points at the extremes of the range). This works well.

To define "peakiness" we adopted an approach intermediate between our old way and the reviewer's proposed way. Basically, our previous method was too liberal, defining peakiness based on the best side of the peak. The reviewer's proposed method was a bit too conservative, defining "peakiness" based on the worst side of the peak (the side with the smaller of ρ_{Left} and $-\rho_{\text{Right}}$). We now use a method that considers both sides of the peak together in a way that is more conservative than our previous method and less severe than the reviewer's suggested method. We now define peakiness as $P = \rho_{\text{Left}} - \rho_{\text{Right}}$, with the additional constraint that the data are not considered consistent with a peak unless both $\rho_{\text{Left}} > 0$ and $\rho_{\text{Right}} < 0$. (We set $P=0$

if $\rho_{\text{Left}} < 0$ or $\rho_{\text{Right}} > 0$). For example, we would consider the case with $\rho_{\text{Left}} = 0.1$ and $\rho_{\text{Right}} = -0.8$ ($P=0.9$) to be more peaky than a case with $\rho_{\text{Left}} = 0.1$ and $\rho_{\text{Right}} = -0.1$ ($P=0.2$). In other words, for a given level of noise, our definition of P favors steeper peaks. We realize this is a bit subjective, but in our view, this is fairer than only considering the worse side of the peak to define P .

For full transparency, we have made available for download the Matlab function that we have made for implementing this statistical test of a noisy peaked relationship. This is pointed out in the Code Availability section of our revised manuscript.

This new way of testing significance was applied in all cases where we report a p value for a noisy peaked relationship between body coupling and population coupling (Figs 4d, 5d, 6c, 6d, 7d, and S5b). We found p values that were usually somewhat larger than our previously reported values, but our conclusions remain solid. In some cases the peak was even more significant.

We note that we have maintained our old way of assessing significance of single recording sessions (Fig 6a, b), because it involves a different hypothesis. Also it was based on a 2nd order polynomial fit. Thus, we presume it was less of a concern to the reviewer.

Reviewer #4 (Remarks to the Author):

The authors have revised their manuscript in response to my additional concerns, significantly improving their statistics and data analysis. This increases my confidence in their findings, and I am now happy to recommend this paper for publication.

One final suggestion (though by no means required) would be to consider modifying the title of the paper. Given that the paper describes a peaked relationship between population coupling and body coupling, the title (while not strictly inaccurate) could potentially lead to confusion.

We thank the reviewer for the supportive comments. We have now further improved the statistical analysis. We would prefer to keep our current title because we often found that the right side of the peak was somewhat more pronounced, which is better reflected in our title.

Reviewers' Comments:

Reviewer #3:

Remarks to the Author:

My main concern still remains because I don't agree with the authors' argument of using an intermediate approach between the authors' old way and my suggested way to define a "peakiness" index.

To prove there is a peak in data, both the left side and right side of the peak should have a significant slope. The authors' method will produce a good peakiness index even if just one side of the data has highly significant slope and the other side is just slightly above 0 (or below 0 depending on which side). Furthermore, I don't think my suggested way is too severe because the same procedure would be applied to shuffled data and the relationship would be defined as significant when the peakiness index of the real data is above 95 percentile of shuffled data. Thus the criterion for a significance level is exactly 0.05, which is standard in our field.

Dear Sachin and Reviewer 3,

Thank you for another, perhaps final round of constructive review. We appreciate the suggestions.

We have now fully adopted the method proposed by the reviewer for testing the statistical significance of a noisy peaked relationship. Our results still hold with this new, stricter definition of “peakiness”.

Below we describe in more detail the specific concern of the reviewer. Our responses are in Arial font, while the reviewer’s comments are in Times New Roman font. Moreover, we note that changes to the manuscript are now highlighted in yellow in our revised manuscript.

Best regards,

Woodrow Shew

Reviewer #3 (Remarks to the Author):

My main concern still remains because I don’t agree with the authors’ argument of using an intermediate approach between the authors’ old way and my suggested way to define a “peakiness” index.

To prove there is a peak in data, both the left side and right side of the peak should have a significant slope. The authors’ method will produce a good peakiness index even if just one side of the data has highly significant slope and the other side is just slightly above 0 (or below 0 depending on which side). Furthermore, I don’t think my suggested way is too severe because the same procedure would be applied to shuffled data and the relationship would be defined as significant when the peakiness index of the real data is above 95 percentile of shuffled data. Thus the criterion for a significance level is exactly 0.05, which is standard in our field.

We understand the reviewer’s concern about our previously proposed approach and we have now implemented the stricter approach suggested by the reviewer. Importantly our main results still hold with statistical significance.

The only results that became statistically insignificant with the new stricter test were for the BC_M vs population coupling for finer time resolution (1/3 of Fig S5b, for 50 and 10 ms). We note that the 100 ms and 250 ms resolution remained significantly ‘peaky’ and the BC_S vs population coupling remained significantly ‘peaky’ for all time resolutions considered.

In addition to changing the method of determining statistical significance for a noisy peak, we made two more small improvements to our work since our last version. First, we noticed that if we include only those recordings with more than 5 single units, this improved our findings. We expect that this improvement is due to the fact that population coupling is not well estimated when based on a small number of single units. Second, we noticed that a few very early recordings (< 2 weeks post implantation surgery) behaved in a qualitatively different manner than our later recordings. This is likely due to an incomplete recovery from surgery. Thus, we

have excluded these early recordings. After these exclusions, our analyzed data still included 1258 units from 143 recording sessions.